# Knowledge Circuits in Pretrained Transformers

**Yunzhi Yao**[1]    **Ningyu Zhang**[1]*    **Zekun Xi**[1]    **Mengru Wang**[1]
**Ziwen Xu**[1]    **Shumin Deng**[2]    **Huajun Chen**[1,3]*
[1] Zhejiang University    [2] National University of Singapore, NUS-NCS Joint Lab, Singapore
[3] Zhejiang Key Laboratory of Big Data Intelligent Computing
{yyztodd,zhangningyu}@zju.edu.cn

## Abstract

The remarkable capabilities of modern large language models are rooted in their vast repositories of knowledge encoded within their parameters, enabling them to perceive the world and engage in reasoning. The inner workings of how these models store knowledge have long been a subject of intense interest and investigation among researchers. To date, most studies have concentrated on isolated components within these models, such as the Multilayer Perceptrons and attention head. In this paper, we delve into the computation graph of the language model to uncover the knowledge circuits that are instrumental in articulating specific knowledge. The experiments, conducted with GPT2 and TinyLLAMA, have allowed us to observe how certain information heads, relation heads, and Multilayer Perceptrons collaboratively encode knowledge within the model. Moreover, we evaluate the impact of current knowledge editing techniques on these knowledge circuits, providing deeper insights into the functioning and constraints of these editing methodologies. Finally, we utilize knowledge circuits to analyze and interpret language model behaviors such as hallucinations and in-context learning. We believe the knowledge circuits hold potential for advancing our understanding of Transformers and guiding the improved design of knowledge editing[1].

## 1 Introduction

*"Knowledge is power, and when embodied in the form of new technical inventions and mechanical discoveries it is the force that drives history."* [1, 2], Bacon's words are vividly re-enacted in the era of Large Language Models (LLMs) [3, 4], as we witness their immense power in reshaping human society and redefining our understanding of machine intelligence. One thing that cannot be denied is that knowledge encapsulated within these models empowers their capabilities in reasoning, perceiving the world, and engaging in human-like communication. Nevertheless, these powerful models are not without their flaws. They still struggle with issues such as hallucinations [5–7], unsafe norms [8, 9], and offensive behaviors [10, 11] and these problems are exacerbated by the enigmatic internal mechanisms of knowledge storage within language models.

Recently, the research community has devoted significant efforts to unraveling the knowledge storage mechanisms of these models. Various studies [12–19] have been conducted to shed light on this intricate process, aiming to enhance our understanding and improve the safety and reliability of language models. The main finding in previous work is that knowledge may primarily stored in the Multilayer Perceptrons (MLPs) of Transformer-based language models. These MLPs function as a key-value neural memory, with knowledge being stored in what are termed "knowledge neurons" (KN). Based on these findings, researchers conduct Knowledge Editing [18, 20] to update the language models' inaccurate facts, bias and unsafe content in their parametric space. Despite the

---

*    Corresponding Author.
[1]Code and data are available in `https://github.com/zjunlp/KnowledgeCircuits`.

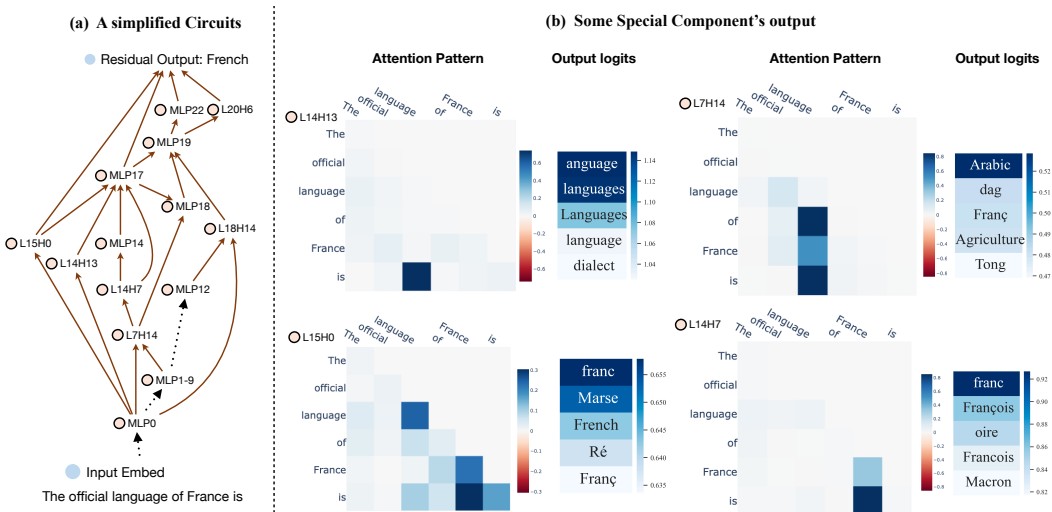

Figure 1: Knowledge circuit obtained from *"The official language of France is French"* in GPT2-Medium. Left: a simplified circuit and the whole circuit is in Figure 9 in Appendix. We use --→ to skip some complex connections between nodes. Here, L15H0 means the first attention head in the 15th layer and MLP12 means the multi-perception layer in the 13th layer. Right: the behavior of several special heads. The matrix on the left is the attention pattern of each attention head and the right heapmap demonstrates the output logits of the hean by mapping to the vocabulary space.

initial success of these methods, there are still limitations, such as poor generalization, severe side effects, and failure to effectively utilize edited knowledge [20, 21], which motivate us to re-think previous approaches for interpreting knowledge storage in language models. Note that previous works treat the knowledge blocks as isolated components following the Restorative Theory [22], often focusing on identifying the specific blocks that store particular knowledge. Several works [23, 24] have proposed that different types of knowledge are often located in the same areas, suggesting that the current KN thesis may be an oversimplification.

To this end, instead of solely pinpointing tiny regions where the knowledge expressed can be localized, we aim to explore the cooperation between different components in Transformers like attention heads, MLPs, and embeddings, to understand how the language model stores and expresses the knowledge. Here, we introduce a new perspective: **Knowledge Circuits**, a critical subgraph in the language model to view the knowledge mechanism of Transformers. Note that Circuit, as a subgraph in the computation graph, has gained ever-growing attention in the mechanistic interpretability field [25]. Previous work [26, 27] has found several important circuits for specific tasks like Indirect Object Identification and Color Object Identification. These tasks necessitate the model to search the preceding context for a matching token and copy it into the next token prediction. In this work, we aim to construct knowledge circuits that require the model to utilize stored knowledge for making predictions. Our goal is to better unveil implicit neural knowledge representations, elucidate the internal mechanisms for knowledge editing, and interpret more complex behaviors of language models. Specifically, we leverage factual recall tasks and conduct experiments across various domains, including factual, social bias, linguistic, and commonsense knowledge. We utilize GPT-2 [28] and TinyLLAMA [29] to explore the potential knowledge representations and utilization mechanisms in these models. As shown in Figure 1 (a), we construct knowledge circuits associated with various expressions of knowledge using the existing knowledge stored in the language model. Through those discovered knowledge circuits, we find many interesting phenomena and conclusions as follows:

**Knowledge circuits unveil implicit neural knowledge representations.** We find that even when the knowledge circuits are used independently, the language model can recall related knowledge with a significant portion of its overall performance, demonstrating the effectiveness of those discovered knowledge representations (circuits). We also delve into specific pieces of knowledge and analyze the information flow within their respective circuits, indicating that the language model tends to aggregate knowledge in the earlier to middle layers and further enhances this information in the later layers. We further uncover several special components (e.g., mover heads and relation heads)

in transferring information to the final token position and capturing relational information from the context (Figure 1 (b)).

**Knowledge circuits elucidate internal mechanisms for knowledge editing.** We conduct experiments to evaluate the impact of current knowledge editing methods on the language models' original knowledge circuits. Empirically, we observe that ROME [18] tends to incorporate edited information primarily at the edited layer. Subsequent mover heads (Appendix B.2) then transport this information to the residual stream of the last token. Conversely, during fine-tuning, the edited token is directly integrated into the language model, exerting a dominant influence on subsequent predictions.

**Knowledge circuits facilitate interpreting language model behaviors.** We further utilize the knowledge circuits to interpret language model behaviors, such as **hallucination** and **in-context learning**. We observe that when hallucination occurs, the language model fails to correctly transfer knowledge to the final token in the earlier layers. This is evident as the knowledge circuit lacks an effective "mover" head, or the mover head selects incorrect information. Additionally, we notice that several new attention heads emerge in the knowledge circuit during in-context learning.

## 2 Background: Circuit Theory

### 2.1 Preliminaries

In the context of neural network interpretability, a circuit can be conceptualized as a human-interpretable subgraph that is dedicated to executing specific tasks within a neural network model [30, 26, 31–33]. When we visualize a neural network model as a connected directed acyclic graph (DAG), denoted as $\mathcal{G}$, the individual nodes represent the various components involved in the forward pass, such as neurons, attention heads, and embeddings. The edges symbolize the interactions between these components, including residual connections, attention mechanisms, and projections. A circuit, represented as $\mathcal{C} \subseteq \mathcal{G}$, emerges as a significant subgraph of $\mathcal{G}$ that is responsible for particular behaviors or functionalities. In this paper, we focus on the Transformer decoder architecture to conduct our experiments. The residual stream of Transformers has been demonstrated to be a valuable tool for mechanistic interpretability in recent works [25, 16]. The Transformer architecture typically starts with token embeddings, followed by a sequence of "residual blocks" and concludes with a token unembedding. Each residual block comprises an attention layer and an MLP layer, both of which "read" their input from the residual stream (via a linear projection) and "write" their output back to the residual stream through an additive projection. We can consider an attention head $A_{l,j}$ (the $j$th attention head in layer $l$) as operating on the residual stream from the previous layer, $R_{l-1}$. Given that $R_0 = I$ (where $I$ represents the input embeddings), we can reinterpret attention head $A_{l,j}$ as processing the cumulative output of all previous attention heads and MLPs and input embedding, treating each node in the previous layers as separate input arguments. Similarly, an MLP node $M_l$ can be seen as operating on the cumulative output of all previous attention heads and MLPs and input embedding, and the output node $O$ operates on the sum of the input embeddings and the outputs of all attention heads and MLPs. The following equations represent the residual connections in the Transformer model, where $R_l$ is the residual stream at layer $l$, and $\text{Input}_l^A$ and $\text{Input}_l^M$ are the inputs to the attention and MLP layers, respectively:

$$R_l = R_{l-1} + \sum_j A_{l,j} + M_l, R_0 = I$$

$$\text{Input}_l^A = I + \sum_{l'<l} \left( M_{l'} + \sum_{j'} A_{l',j'} \right)$$

$$\text{Input}_l^M = I + \sum_{l'<l} M_{i'} + \sum_{l'\leq i} \sum_{j'} A_{l',j'}$$

The computational graph $\mathcal{G}$ of the Transformer represents the interactions between attention heads and MLPs. The nodes in $\mathcal{G}$ encompass the input embedding $I$, attention heads $A_{l,j}$, MLPs $M_l$, and the output node $O$, denoted as $N = \{I, A_{l,j}, M_l, O\}$. The edges in the model represent the connections between these nodes, $E = \{(n_x, n_y), n_x, n_y \in N\}$. A circuit $\mathcal{C}$ is meticulously constructed to govern specific behaviors within the model, comprising a selection of nodes $N_{\mathcal{C}}$ and edges $E_{\mathcal{C}}$ that are critical to the successful execution of the tasks at hand, expressed as $\mathcal{C} =< N_{\mathcal{C}}, E_{\mathcal{C}} >$.

## 2.2 Circuit Discovery

To identify circuits within a language model, a key approach is to examine the model's casual mediation by systematically altering the model's edges and nodes to observe the effects on performance [32, 34, 35]. The underlying principle is that critical edges or nodes are those whose removal results in a notable decline in the model's predictive capabilities. Since the edges in the model's computational graph represent the dependencies between nodes, we can simulate the absence of a particular node-to-node dependency by ablating an edge in the graph. For example, ablating an edge from $A_{i',j'}$ to $A_{i,j}$ involves replacing the contribution of $A_{i',j'}$ in the input to attention head $A_{i,j}$ with zero (in the case of zero ablation) or with the mean value of head $A_{i',j'}$ (in the case of mean ablation). The process of identifying critical edges or nodes through ablation can be broken down into the following steps: i) Overwrite the value of the edge $(n_x, n_y)$ with a corrupted value (either zero or mean ablation), ii) Perform a forward pass through the model with the altered graph, iii) Compare the output values of the modified model with those of the original model using a chosen metric $S$ (Details in Eq.1 ). If the performance change is below a predefined threshold $\tau$, we can consider the edge non-critical and remove it to obtain a new subgraph $\mathcal{G}/(n_x, n_y)$. In addition to ablation-based methods, recent works have also explored the use of sparse auto-encoders [36, 37] to identify circuits within language models. This approach involves training an auto-encoder to learn a sparse representation of the model's internal structure, which can help reveal the underlying circuitry responsible for specific behaviors or functionalities.

# 3 Knowledge Circuits Discovery in Transformers

## 3.1 Knowledge Circuits Construction

Unlike previous work [12, 18], which managed to find out the specific areas that store knowledge, we pay extra heed to the information flow that activates subsequent knowledge for answering questions. Similar to [38, 26], we write language model as a graph consisting of the input, the output, attention heads, and MLPs by considering a "residual rewrite" of the model's computational structure. For example, this residual rewrite gives us a nearly-dense graph in GPT2-medium: one between every pair of (attention head, MLP, input, and output) nodes, except for attention heads in the same layer, which do not communicate with each other. In our paper, we concentrate on the task of answering factual open-domain questions, where the goal is to predict a target entity $o$ given a subject-relation pair $(s, r)$. A knowledge triplet $k = (s, r, o)$ is often presented to the model in the form of a natural language prompt for next token prediction (e.g., *"The official language of France is ____"*). The model $\mathcal{G}$ is expected to generate the target entity, which is consistent with the language model's pretraining format. To identify the circuit that is critical for predicting the target entity $o$ for a given subject-relation pair $(s, r)$, we ablate each special edge $e_i = (n_x, n_y)$ in the computation graph $\mathcal{G}$. We then measure the impact of ablating the edge (**zero ablation** in our implementation) on the model's performance using the MatchNLL loss [32] for the target $o$:

$$S(e_i) = \log(\mathcal{G}(o|(s, r))) - \log(\mathcal{G}/e_i(o|(s, r))) \tag{1}$$

If the score $S(e_i)$ is less than the predefined threshold $\tau$, we consider the edge to be non-critical and remove it from the computation graph, updating the temporary circuit $\mathcal{C}_{temp} \leftarrow \mathcal{G}/e_i$. We first sort the graph by topological rank following Conmy et al. [32] and traverse all edges in this manner, We derive a circuit $\mathcal{C}_k$ that contributes to representing the knowledge necessary to answer the factual question:

$$\mathcal{C}_k = < N_k, E_k > \tag{2}$$

Here, $\mathcal{C}_k$ is the circuit for the knowledge triplet $k$, consisting of the nodes $N_k$ and edges $E_k$ that are essential for predicting the target entity $o$ given the subject-relation pair $(s, r)$.

## 3.2 Knowledge Circuits Information Analysis

Once we have identified the knowledge circuit, we delve deeper into the specific roles and behaviors of each node and edge within the computation graph. Our goal is to comprehend the processing and contribution of each node $n_i$ to the functionality of the circuit. Drawing on the methodologies of previous studies [16, 39, 40], we begin by applying layer normalization to the output of each node $n_i$ and then map it into the embedding space. This is achieved by multiplying the layer-normalized output by the unembedding matrix ($\mathbf{W}_U$) of the language model: $\mathbf{W}_U \text{LN}(n_i)$. This transformation allows

us to inspect how each component writes information to the circuit and how it influences subsequent computational steps. By understanding the nodes' behavior in the circuit, we can better comprehend the circuit's structure and the key points where information is aggregated and disseminated.

Table 1: **Hit@10** of the Original and Circuit Standalone performance of knowledge circuit in GPT2-Medium. **The result for $D_{val}$ being 1.0 indicates that we select the knowledge for which the model provides the correct answer to build the circuit.**

| Type | Knowledge | #Edge | $D_{val}$ | | $D_{test}$ | | |
|---|---|---|---|---|---|---|---|
| | | | Original($\mathcal{G}$) | Circuit($\mathcal{C}$) | Original($\mathcal{G}$) | Random | Circuit($\mathcal{C}$) |
| Linguistic | Adj Antonym | 573 | 0.80 | 1.00 ↑ | 0.00 | 0.00 | 0.40 ↑ |
| | word first letter | 432 | 1.00 | 0.88 | 0.36 | 0.00 | 0.16 |
| | word last letter | 230 | 1.00 | 0.72 | 0.76 | 0.00 | 0.76 |
| Commonsense | object superclass | 102 | 1.00 | 0.68 | 0.64 | 0.00 | 0.52 |
| | fruit inside color | 433 | 1.00 | 0.20 | 0.93 | 0.00 | 0.13 |
| | work location | 422 | 1.00 | 0.70 | 0.10 | 0.00 | 0.10 |
| Factual | Capital City | 451 | 1.00 | 1.00 | 0.00 | 0.00 | 0.00 |
| | Landmark country | 278 | 1.00 | 0.60 | 0.16 | 0.00 | 0.36 ↑ |
| | Country Language | 329 | 1.00 | 1.00 | 0.16 | 0.00 | 0.75 ↑ |
| | Person Native Language | 92 | 1.00 | 0.76 | 0.50 | 0.00 | 0.76 ↑ |
| Bias | name religion | 423 | 1.00 | 0.50 | 0.42 | 0.00 | 0.42 |
| | occupation age | 413 | 1.00 | 1.00 | 1.00 | 0.00 | 1.00 |
| | occupation gender | 226 | 1.00 | 0.66 | 1.00 | 0.00 | 0.66 |
| | name birthplace | 276 | 1.00 | 0.57 | 0.07 | 0.00 | 0.57 ↑ |
| **Avg** | | | 0.98 | 0.73 | 0.44 | 0.00 | 0.47 ↑ |

## 3.3 Knowledge Circuits Experimental Settings

**Implementations.** We conduct experiments on GPT-style models, including GPT-2 medium and large. We also conduct primary experiments on TinyLLaMA [29] to validate the effectiveness of different architectures. We utilize the Automated Circuit Discovery [32] toolkit to build a circuit as an initiative of our analysis and leverage transformer lens [41] to further analyze the results. Specifically, we simply employ the MatchNLL [32] as the metric to detect the effect of the given node and edge and use **zero ablation** to knock out the specific computation node in the model's computation graph.

**Metrics.** A discovered knowledge circuit is deemed an accurate representation of a specific area within the transformer's knowledge storage, thus, it should be capable of representing the knowledge independently. Following [32], we leverage the completeness of a circuit, which refers to its ability to independently reproduce the behavior or predictions of the full model for the relevant tasks. This property is assessed by examining whether the identified subgraph corresponds to the underlying algorithm implemented by the neural network. To evaluate completeness, we first construct the circuit using the validation data $D_{val}$ for a specific knowledge type and then test its performance on the test split $D_{test}$ in isolation. By doing so, we can observe any changes in performance compared to the original model. We use the **Hit@10** metric to measure the rank of the target entity $o$ among the top 10 predicted tokens:

$$\texttt{Hit@10} = \frac{1}{|V|} \sum_{i=1}^{|V|} \text{I}\left(\text{rank}_o \leq 10\right) \tag{3}$$

Here, $|V|$ represents vocabulary size, and $\text{rank}_o$ is the rank of the target entity $o$ in predictions.

**Dataset.** In this work, we focus on the **knowledge that is already stored in the language model**. We utilize the dataset provided by LRE [42] and consider different kinds of knowledge, including linguistic, commonsense, fact, and bias. We evaluate whether the knowledge is present in the language model's parameters under zero-shot settings using the **Hit@10** metric to sample knowledge from the validation set, which is used to construct the knowledge circuit. The data statistics are in Appendix A.

## 4 Knowledge Circuits Unveil Implicit Neural Knowledge Representations

**Knowledge Circuits Evaluation.** We report the results of GPT2-Medium in Table 1, which indicates that with only less than 10% of the original knowledge circuit's subgraph, the model can

maintain over 70% of its original performance. Additionally, we compute the random circuits by randomly deciding whether the edge should be removed and making sure the graph is connected. The random circuit is the same size as the circuit we discovered using our method. From the table, we can see that the random circuit failed to maintain the model's performance, which further enhanced the robustness and efficacy of our methods. One of the most fascinating observations is the **performance improvement seen on several test datasets**. For instance, the *Landmark-country* relation metric increases from 0.16 to 0.36. This suggests that the discovered knowledge circuits may encapsulate the relevant knowledge, and the model's performance on these tasks could have been hindered by noise from other components. We proceed to analyze the layer distribution of the original model $\mathcal{G}$ to understand the average percentage of nodes that are activated within the circuit for different knowledge domains. From Figure 2, we observe that attention and MLPs are more active in the lower layers of the network, where the language model processes the input and extracts general information. To gain a more comprehensive view of the information processing, we compute the average $\text{rank}_o$ change of the target token in the $D_{val}$ across the layers and report the results in Figure 7. This analysis reveals the phenomenon of early decoding [40], suggesting that by the middle to the latest layers, the target entity is already present in the residual stream, and the subsequent layers in the Transformer are designed to increase the probability of the current token (See discussion in the running example).

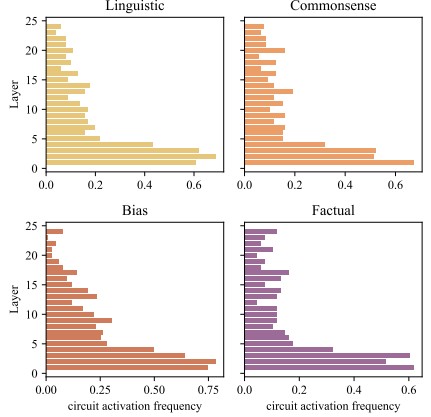

Figure 2: The activated circuit component distributions in Layers in GPT2-Medium.

**Special Components in Knowledge Circuits.** From the discovered knowledge circuits, we can find several important attention heads that demonstrate specific behavior, including the **mover head** [31], **relation head** [17, 43] and **mixture head** [17, 43] (more definitions in Appendix B.2). Mover Head [31, 27] focuses on the last token of the context and attends to the subject token, functioning as a mover to transfer information, while Relation Head [17] attends to the relation token in the context and produces some relation-related tokens that would guide the behavior of the following components. We think that these components would be accumulated by the MLP in the model, and the behavior of these special heads will be discussed in the running example part. We list some of these special components in Table 4 in Appendix. The different attention heads are responsible for expressing specific types of knowledge and may be activated by different facts. In our experiments with GPT-2 Medium and GPT-2 Large, we find that knowledge is distributed across several layers' attention heads and MLP matrices, suggesting that the target knowledge appears to have been accumulated throughout the GPT-2 model. Conversely, in TinyLLAMA, the special components are more concentrated. As depicted in Figure 7, the rank of the target entity in TinyLLAMA experiences a sharp decline around several layers, whereas in the GPT2 model, the decline is more gradual. We hypothesize that this discrepancy may be attributed to the model's knowledge capacity [44] and warrants further investigation.

**A Running Example of Knowledge Circuit.** We present a case and analyze the specific behaviors of components within the identified knowledge circuits. In particular, we find some special attention heads in the model such as the mover head and the relation head. We demonstrate the function of these heads in Figure 6 by ablating them from the circuit. Taking the factual knowledge *"The official language of France is French"* as an example, we visualize the knowledge circuit in Figure 1. To express the information flow within the model more effectively, we have plotted the rank and probability of the target entity $o$ at each layer when it is mapped into the embedding space, in Figure 3. From this figure, we can see that after MLP 17, the target knowledge emerges as the top token in the residual stream, and after that layer, it undergoes an increased probability. The edges connected to MLP17 are (L14H13 → MLP17), (L14H7 → MLP17), and (L15H0 → MLP17) . Here, the L14H13 is a relation head that focuses on the relation token in the context. The output of this head is relation-related tokens such as *"language"* and *"Language"*. The attention head L14H7 is a mover head that moves the information from the subject position *"France"* to the last token. Previous work [31, 19] has introduced this mover head as an *argument parser*, which moves *"France"* to the last token, and the subsequent MLP conducts a function application to map *"France"* to *"French"*. An intriguing

observation is that we can find the output of this head already contains the target answer entity, which significantly contributes to the final output (L14H7 → Output). Also, we see the probability of the subject token in Figure 3 at the last token is nearly zero across these layers. Hence, instead of the *argument parser* function, we consider this mover head as an *extract head* proposed by Geva et al. [13], which aims to extract the related-information from the subject token's position. In the subsequent knowledge editing experiments, we can observe changes in the behavior of these types of heads.

Additionally, instead of extraction in the later layers proposed by Geva et al. [13], we notice a gradual decrease in rank across all early-to-middle layers. The MLP17 combines information from previous tokens and integrates this information to prioritize the target token at the top rank.

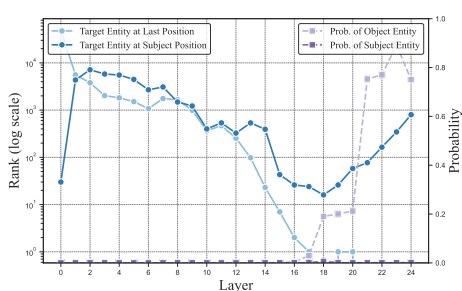

Interestingly, upon tracing the information flow to L14H7, we discovered that it is predominantly activated by L7H14, a relation head, and its output features several language tokens, such as *"Arabic"*. We hypothesize that L7H14 may function as a signaling mechanism to activate the associated mover head, but this hypothesis necessitates further investigation to be confirmed. After MLP17, several attention heads, such as L18H14 (a relation head) and L20H6 (a mover head), collaborated to further enhance the final prediction of the target entity.

Figure 3: The rank and probability of the target entity $o$ at both the last subject token and the last token position when unembedding the intermediate layer's output for the fact *"The official language of France is French"*.

## 5 Knowledge Circuits Elucidate Internal Mechanisms for Knowledge Editing

In this section, our objective is to evaluate the impact of previous knowledge editing methods and validate the effectiveness of knowledge circuits. We aim to understand why these methods may fail in certain cases and settings, which can also help deepen the understanding of the knowledge circuit.

**Single Factual Knowledge Editing.** Here, we adopt the ROME method [18] and FT-M [24], which aim to edit the MLP layers in the language model. The most important hyper-parameter in knowledge editing is the layer, as the same method's performance varies significantly via the layers. Here, we evaluate the performance of different editing layers and their effectiveness. We compare the knowledge circuits computed by the edited model with the original one, and we present results in Figure 4 and report details in Appendix D. As discussed in the previous part, the early-to-middle layers are the main part of aggregating the target entity $o$ to the top rank. In the original model, the probability of the target entity *"Intel"* is nearly zero, and the model fails to elevate it to the top rank in the vocabulary. Editing the model with ROME and FT-M both give us the correct answer but we can view different scenarios for their knowledge circuits. For **ROME**, as the correct information is added to the subject position, we can recognize a **behavior of the Mover Head shifts from copying to extracting the edited information from the subject position**. This information gradually aggregates through the subsequent layers, and by layer 15, *"Intel"* emerges as the top-ranked entity with its probability increasing significantly. Specially, before editing, the mover head L15H3 attends to the *"controller"* token and returns *"controller"* as the output, while in the edited model, the attention head's output moves to the *"Intel"*, which means the model gains the information at the subject space. For FT-M, the edited model tends to directly write the knowledge into the specific component, which would greatly dominate the following component in the model. As shown in Figure 4, the output logits in MLP-0 for *"Intel"* are more than 10, and it emerges as the top rank in the residual stream directly. This phenomenon can be found in different knowledge types and layers and we report results in Appendix D.2. However, the added knowledge may have the risk of influencing unrelated knowledge. When we test another fact *"Windows server"*, the model still tends to give us the *"Intel"* answer, demonstrating the overfitting problem. This finding supports previous analysis regarding the correlation between localization and editing [45], suggesting that edits may not alter the storage but merely add signals into the knowledge circuits.

**Multi-hop Factual Knowledge Editing.** Multi-hop knowledge editing poses a challenging scenario [20, 21, 46], wherein we edit the model with new knowledge, yet the model struggles to perform reasoning using the edited information. We analyze multi-hop questions in language models [47, 48]

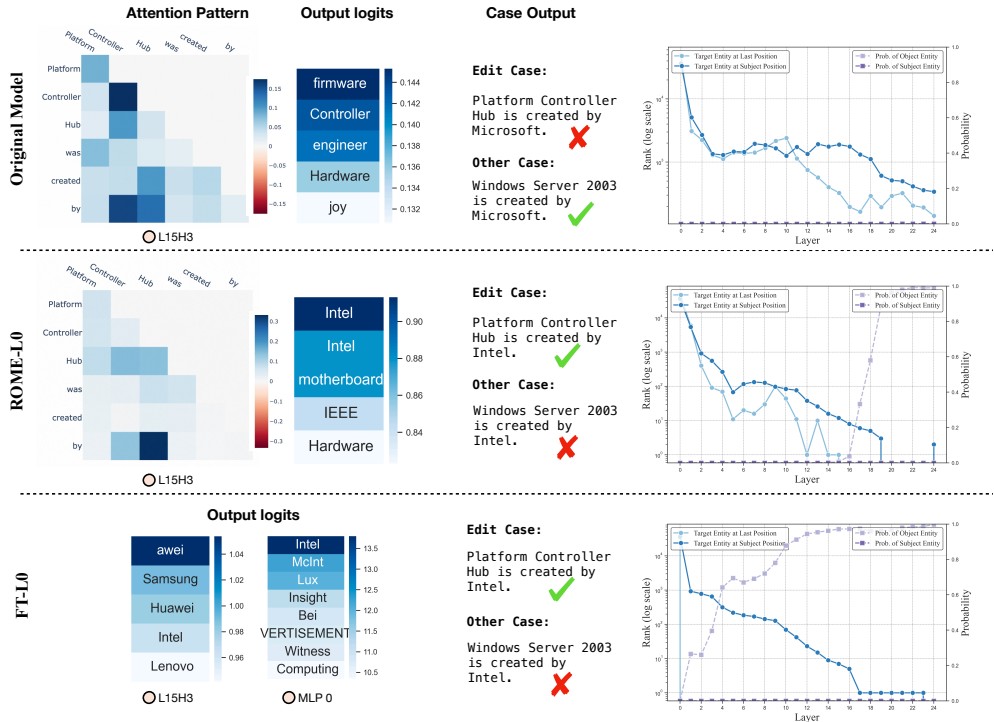

Figure 4: Different behaviors when we edit the language model. In the original model, we can see the mover head L15H3 actually move the original token *"Controller"* and other information, while for ROME, we observe the mover head select the correct information *"Intel"*, which means ROME **successfully added the *"Intel"* to model**. For the FT layer-0 editing, we can find this method **directly write the edited knowledge into edited component**. However, we find these two editing methods would affect other unrelated input *"Windows server is created by?"*

to understand why current editing methods fail in these scenarios. For instance, given the fact (Thierry Mugle, *"home country"*, France), we edit the fact to another country, such as (Thierry Mugle, *"home country"*, France → China). We then assess the model's performance on questions based on the edited knowledge, including *"The official currency of the home country of Thierry Mugle is"* and *"The capital city of the home country of Thierry Mugle is"*. While the unedited model could correctly answer these questions, we observe that the edited model would provide the answer *"China"* for subsequent hop reasoning. We find that the mover head in the original multi-hop reasoning circuit initially extracts the second-hop answer but, after editing, extracts *"China"*, demonstrating that the edited information dominantly saturates and influences the circuit. Furthermore, we observe an intriguing phenomenon: **even in the original model's multi-hop reasoning settings, it would directly provide the answer if we remove the context of the first-hop texts** (Details in Appendix C.1). This further confirms the findings that the model relies on relational and subject-related information, regardless of grammatical adherence.

## 6 Knowledge Circuits Facilitate Interpreting Language Model Behaviors

In this Section, our aim is to validate whether the identified knowledge circuits are actually utilized by the model when it employs knowledge. To address this, as shown in Figure 5, we investigate three phenomena: hallucination, in-context learning, and reverse relations (Details in Appendix C.3).

**Factual Hallucination.** If the knowledge is stored and expressed by the circuit we discovered, we aim to discover what happened when the model gave us the incorrect answer. We focus on factual hallucinations, which occur when the model provides an incorrect target entity for a given subject $s$ and relation $r$. In our experiments (Figure 5 and Appendix C.2), we observe that the model fails to move the correct knowledge to the final token in the earlier layers. This failure is evident as the

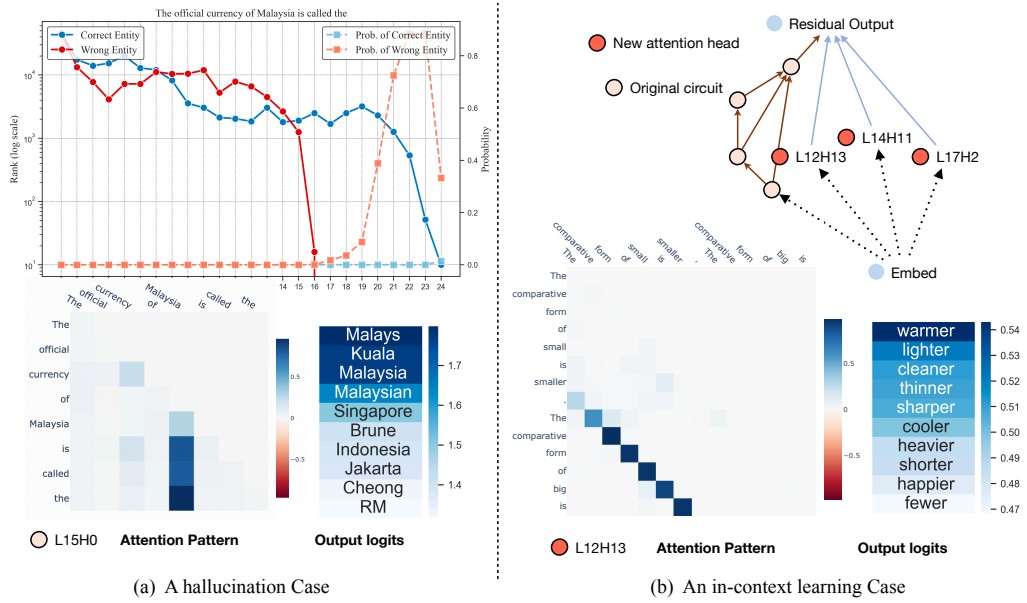

(a) A hallucination Case  (b) An in-context learning Case

Figure 5: Left: fact hallucination case *"The official currency of Malaysia is called"*, we observe that, **at layer 15, the Mover Head selects incorrect information**. Right: In-context learning case, we notice that **some new heads focusing on the demonstration appear in the knowledge circuit**.

circuit lacks an effective mover head or the mover head selects incorrect information. For instance, in the prompt *"The official currency of Malaysia is called"*, both the correct answer *"Ringgit"* and the incorrect one *"Malaysian"* are accumulated before layer 15. However, at layer 16, the mover head L15H10 extracts the erroneous information. Despite a rank drop of the true one in layers 20–22, this is insufficient to correct the previous mistake.

**In-Context Learning.** Despite storing a vast amount of knowledge, a language model may still provide incorrect answers. However, with demonstrations or examples (based on RAG [49]), it can quickly generate correct responses. To this end, we focus on the scenario where the model initially provides an incorrect answer but can then produce the correct response upon receiving the appropriate demonstration. We consider the original knowledge circuit and introduce a new knowledge circuit based on the demonstration. Our analysis reveals that, compared to the zero-shot knowledge circuit, several new attention heads appear in the computation graph when the demonstration is incorporated. We show the behavior of these attention heads in Figure 5. We can see these heads mainly focus on the demonstration's context: *"The comparative form of small is smaller"* and works as the Induction Head [50] that look back over the sequence for previous instances of the current token and find the token that came after it last time. To better view the function of these heads, we conduct experiments by ablating the newly appeared attention head in the ICL circuit in Table 2. We find that compared to the randomly selected attention head by ablating this attention, the probability drops significantly in the prediction, demonstrating the importance of these identified attention heads. These aligned with previous work where Todd et al. [51] have identified a concept known as the Function Vector, which represents the average of some key attention heads and provides the task learning ability.

# 7 Related Work

**Knowledge Mechanism of Transformers.** How the language model stores and utilizes knowledge is an ongoing research topic. Previous works find that the MLP in Transformers works as a key-value memory and stores enormous knowledge [12, 15, 14, 18]. As to the relation between entities, Hernandez et al. [42] observes that facts can be decoded linearly from the enriched residual stream of the subject by mapping the subject entity to the object entity. Instead of viewing the knowledge storage in isolation, Geva et al. [13], Lv et al. [31], Yu and Ananiadou [16] find the knowledge is accumulated during the layers. Regarding knowledge analysis, Bayazit et al. [52] also attempts to discover critical knowledge in language models. However, they only consider several layers in the

Table 2: Performance change via ablating the newly appeared attention heads in the ICL circuit and random heads.

| | Knowledge | Origin Model | Ablating extra head | Ablating random head |
|---|---|---|---|---|
| Linguistic | adj_comparative | 62.24 | 32.55 | 58.18 |
| Commonsense | word_sentiment | 89.02 | 55.50 | 88.61 |
| | substance_phase | 78.74 | 52.85 | 71.24 |
| Bias | occupation_gender | 86.97 | 59.54 | 86.54 |
| Factual | person_occupation | 35.17 | 23.27 | 31.60 |

model and use the pruning method, which may overlook the connections between components. More related works can be found in Appendix E.1.

**Manipulate Language Models.** Recently, many works aim to manipulate the language models to make the model aligned with world knowledge or social value norms, such as knowledge editing [20, 24], machine unlearning [53, 54] and detoxification [55, 56]. Most of these works are elicited by previous knowledge mechanism findings such as knowledge neuron [57]. They modify the MLP in the LLM [18, 12] to change the model's behavior based on specific factual knowledge. However, recent works [58, 59] demonstrate the pivotal role of the attention part in knowledge representation. Hase et al. [45] also observe that the performance of editing within a layer may not reliably pinpoint the location of the fact. In this paper, we try to manipulate specific knowledge of language models via knowledge circuits, including both MLP and attention components across different layers.

# 8    Conclusion

In this paper, we present a new perspective on knowledge storage based on circuit theory and conduct a preliminary analysis to demonstrate its effectiveness. We found that knowledge circuits in the model are not only responsible for expressing knowledge but can also guide behavior in different settings. We hope these findings can advance our understanding of the knowledge mechanisms of language models and provide insights for better designing and editing language models, enhancing knowledge, and improving reasoning to enhance factuality and alleviate hallucinations.

## Limitations and Broader Impacts

In this work, we employ the causal mediation method to automatically construct circuits tailored to specific knowledge domains. However, this circuit discovery-based patching approach is time-intensive. Contemporary research efforts have introduced more efficient methodologies for modeling information flow [60–62]. Additionally, alternative techniques for discovering circuits through mask training [63, 64] and Sparse Auto-Encoders [65, 36] have been proposed, highlighting diverse facets of circuit behavior within large language models (LLMs). We posit that the field of knowledge circuit discovery holds significant potential for advancement. Furthermore, recent studies [66] have developed 'circuit breakers' to manage representations associated with potentially harmful outputs. We hope that our approach can contribute to ensuring the safety and privacy of information, thereby fostering the development of trustworthy AI.

## Acknowledgments

This work was supported by the National Natural Science Foundation of China (No. 62206246, No. NSFCU23B2055, No. NSFCU19B2027), the Fundamental Research Funds for the Central Universities (226-2023-00138), Zhejiang Provincial Natural Science Foundation of China (No. LGG22F030011), Yongjiang Talent Introduction Programme (2021A-156-G), CIPSC-SMP-Zhipu Large Model Cross-Disciplinary Fund, Ningbo Science and Technology Special Projects under Grant No. 2023Z212, Information Technology Center and State Key Lab of CAD&CG, Zhejiang University, NUS-NCS Joint Laboratory (A-0008542-00-00), and the Ministry of Education, Singapore, under the Academic Research Fund Tier 1 (FY2023) (Grant A-8001996-00-00). We gratefully acknowledge the support of Zhejiang University Education Foundation Qizhen Scholar Foundation.

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

# Appendix

## A Implementation Details

**Hyper-parameter.** The primary hyperparameter for constructing a circuit is the threshold $\tau$ used to detect performance drops. Setting $\tau$ too high may result in an incomplete circuit, while setting it too low can introduce numerous unnecessary nodes. In our experiment, we test $\tau$ values from the set {0.02, 0.01, 0.005} to determine the appropriate circuit size for different types of knowledge.

**Implementation.** We utilize ACDC[2] to construct circuits that encode specific knowledge representations. Additionally, we re-implement the code to compute the relevant dataset and impact metrics for the knowledge used. Since TinyLLAMA[3] incorporates the Grouped Query Attention Mechanism [67], we interleave and repeat the key and value pairs to analyze the specific behavior of each attention head. We use the NVIDIA-A800 (40GB) to conduct our experiments. It took about 1-2 days to compute the circuit for the knowledge type in GPT2-medium.

**Dataset Details.** All the data used in our paper is sourced from Hernandez et al. [42], with the detailed information provided in Table 6. In the original setting, they use the data for few-shot settings, but in our experiments, we consider zero-shot knowledge storage, so here we sample the data using the `Hit@10` to detect whether the model understands knowledge for the given prompt based on the $s$ and $o$. We sample the test set in a 1:1 ratio with the validation set to ensure a balanced evaluation.

| Category | # Rel. | # Examples | # GPT-2 Corr. |
|---|---|---|---|
| Factual | 26 | 9,696 | 4,721 |
| Commonsense | 8 | 374 | 240 |
| Linguistic | 6 | 806 | 483 |
| Bias | 7 | 213 | 149 |

Table 3: Information about the dataset. Table is borrowed from Hernandez et al. [42]

## B More Experiment Results

### B.1 Rank Change Across Layers

To gain a clearer understanding of the knowledge aggregation phenomenon, we compute the rank of the target entity $\text{rank}_o$ within the vocabulary space $|V|$ at the output of each layer. As depicted in Figure 7, we observe that the model initially elevates the target entity to the top ranks of the vocabulary. Once the entity reaches the top of the vocabulary, subsequent layers continue to enhance its probability mass. These findings corroborate previous work by [68–70], who note the substantial difference in logit entropy between layers, contributing to the model's improved prediction for the target entity. In our work, we aim to delve deeper into how the model, as well as specific components within it, give rise to these behaviors.

### B.2 Special Components in Knowledge Circuit

When zooming into the discovered circuit, we can find several kinds of special attention heads, or MLPs, in the model that play a pivotal role in the final prediction, similar to what previous research has indicated [31]. Apart from mover head and relation head, there is another kind of head named **Mix Head** [17, 43] which would focus on both the relation token and the subject tokens. In our experiments, we found these heads usually work similarly to the mover head. In particular, mover heads contribute more to the subject-specific information, as ablating them would increase other relation-related tokens' probability. Moreover, if we ablate the relation head, we can find the model tends to generate some meaningless tokens instead of the relation information. From Figure 3 6, we can find ablating the mover head would increase the probability of "Italian", "English" and "Spanish", which are not subject-related. While ablating the relation head would lead to the increase of some meaningless words "a", and "that", which are not relation-related.

Lv et al. [31] posits that these mover heads are responsible for extracting the "argument" from the context and passing it for further processing, such as function application. Consequently, Lv et al. [31], Merullo et al. [19] suggests that the MLP within the language model performs a function

---

[2]https://github.com/ArthurConmy/Automatic-Circuit-Discovery
[3]Checkpoint: https://huggingface.co/TinyLlama/TinyLlama-1.1B-intermediate-step-1431k-3T

**Top 10 Token Output Probability**

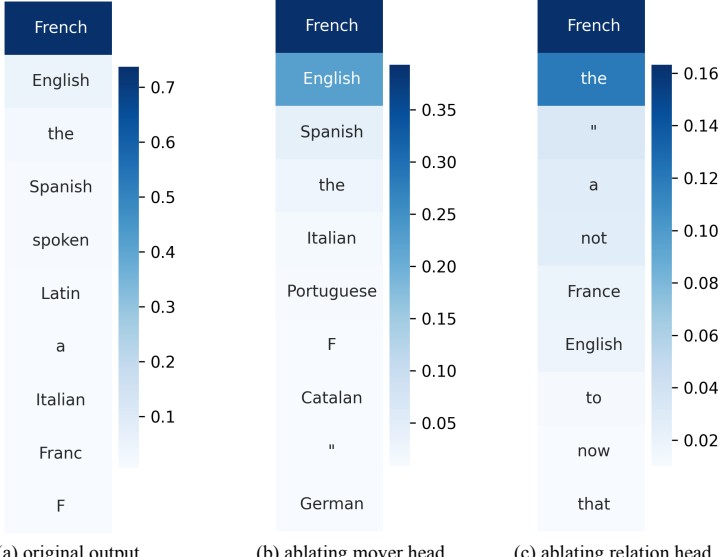

(a) original output     (b) ablating mover head     (c) ablating relation head

Figure 6: The output of the model. Ablating the mover head would increase the probability of "Italian", "English" and "Spanish", which are not subject-related. While ablating the relation head would lead to the increase of some meaningless words "a", "that", which are not relation-related.

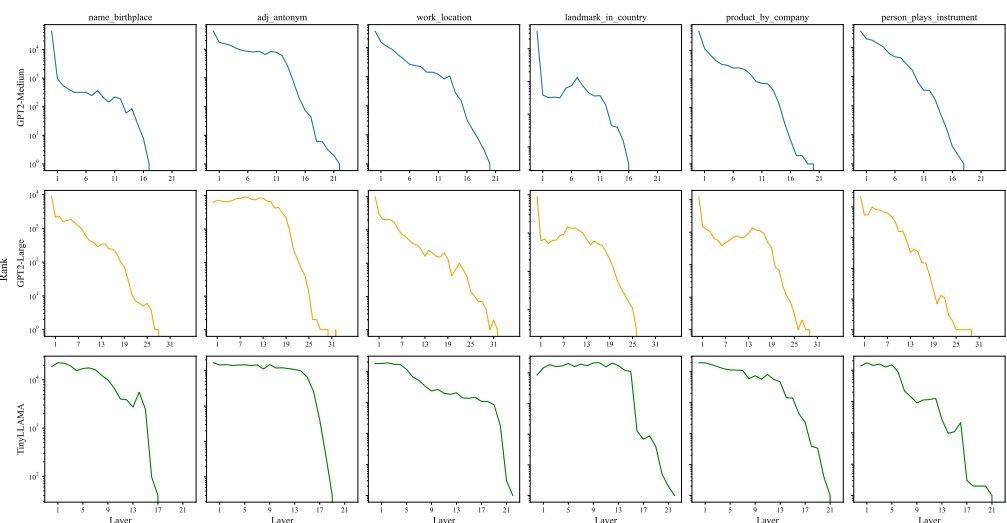

Figure 7: The average rank of the target entity $o$ for the $D_{val}$ in the vocabulary when mapping the output of each layer in the model to the embedding space. We can find that the in GPT2-Medium and GPT2-Large, the model would get the knowledge at middle-to-later layers. While for TinyLLAMA, the layer may be more later.

application that transforms the subject into an object, with the subject's probability ranking higher than the object's. However, these findings are limited to specific cases, as the studies only examine two related tasks, such as capital city identification or color objection. Our analysis suggests that these conclusions may not be universally applicable to all knowledge domains and require further investigation. When we examine the ranks of subject and object entities, we rarely observe an overwhelming superiority at the last token position of the subject token and usually, the probability of the subject token at the last position is uniformly low. In our experiment, we view the model as a collaboration between different nodes in the identified circuit. They perform as different kinds of

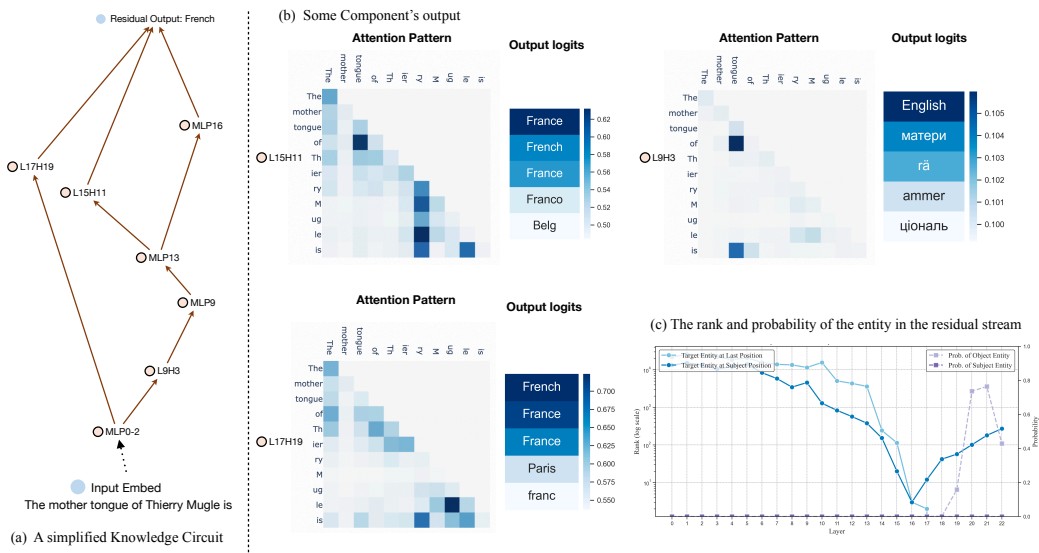

Figure 8: A simplified knowledge circuit found in TinyLLAMA for the knowledge *"The mother tongue of Thierry Mugler is French"*.

attention heads, like *mover head* and *relation head*. We list some heads that are responsible for the storage of different kinds of knowledge and relations in Table 4.

**Component Reuse Phenomenon.** Merullo et al. [27] have identified shared circuits for the IOI task and the Colored Objects task. We also observe this phenomenon in the factual recall task. As depicted in Table 4, we can observe that for related relations such as "city_in_country", "name_birth_place", and "country_language", their circuits include both L21H12, which stores and maps country-related information. Additionally, we found that some *relation heads* are activated by different relations. For instance, in our experiments, the head 'L7H14' appears in the circuits of both "official_language" and "official_currency". We speculate that these reused heads, rather than task-specific heads, can be considered topic heads, as proposed by [71, 60]. We believe that further investigation into this distinction is warranted in future research.

Table 4: Special component behaviour in circuits as task-specific head. Find more results in Appendix

| Model | Type | Fact | Critical Component in Circuit |
|---|---|---|---|
| GPT2-Medium | Linguistic
Factual
Commonsense
Bias | Antonym
city country
work location
name country | L17H2, L18H1, L13H12, L13H8
L21H12, L16H2
L19H15, L14H4, L13H3
L16H6, L21H12 |
| GPT2-Large | Linguistic
Factual
Commonsense
Bias | Antonym
company hq
work location
name country | L25H5, L24H16, L19H13, L18H8
L30H6, L25H13
L18H13, L28H18, L30H5
L21H19, L29H2 |
| TinyLLAMA | Linguistic
Factual
Commonsense
Bias | Verb past tense
Landmark country
Fruit Inside Color
name country | L17H0, MLP20
L15H11, L17H19, MLP18
L18H25, MLP18
L15H11, MLP17 |

## B.3 A Case on TinyLLAMA

In this part, we demonstrate one circuit we found in the TinyLAMA in Figure 8. Actually, in TinyLLAMA, the attention heads bearing specific behaviors in the later layers is usually less than

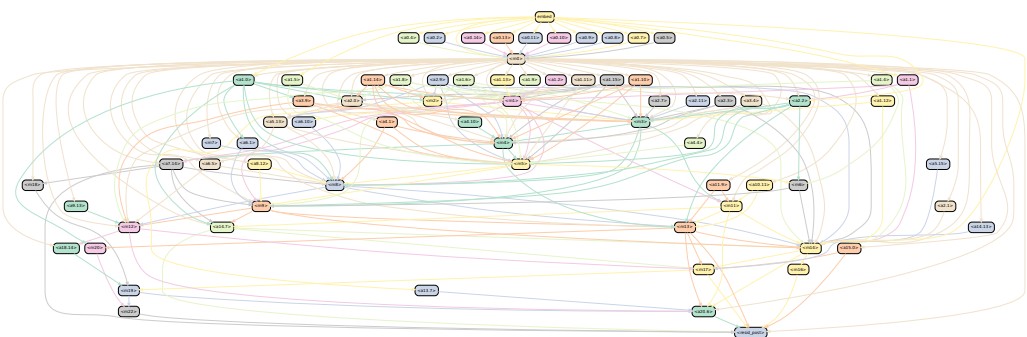

Figure 9: The knowledge circuit from the *"The official language of France is French"* in GPT2-Medium.

GPT2. We can also view some mover heads and relation heads in the circuit that would generate the target token as the output, such as L15H3 and L17H19.

## C   More Circuit Utilization Analysis

While previous work on knowledge storage suggests that knowledge may be localized in specific areas of the model, it is essential to ascertain whether the model actively employs this knowledge when encountering related contexts or if it relies on shortcuts.

### C.1   Multi-hop Factual Knowledge Editing

We consider scenarios where the model makes a correct prediction for all the multi-hop questions and single-hop questions. We found a circuit reuse phenomenon in the one-hop and multi-hop knowledge circuits. Here, we first compute the proportion of the nodes $N_{single}$ in the single-hop circuit $C_{single}$ that appears in $N_{multiple}$ the set of nodes in the multi-hop circuit $C_{multiple}$ .

$$Hit_{node} = \frac{|N_{multiple}| \cap |N_{single}|}{|N_{single}|} \qquad (4)$$

Actually, there are two ways for the model to conduct multi-hop reasoning. As shown in the figure, the model can also answer the question by combining the two hop relations together (in the given case, combine "hometown" and "language" as "mother tongue") If the model is capable of combining two-hop relations in a more integrated or semantic way, such as inferring that the "mother tongue" is the language of one's hometown, this suggests a more complex reasoning process that goes beyond the simple overlap of nodes and edges. To capture this kind of reasoning, we assess the model's ability to integrate information from different hops. $R_{integrated}$ as the set of integrated relations (new paths created by combining information from different hops)

We observe an overlap of these circuit nodes, indicating that the language model utilizes a large portion of the nodes in the original single hop's circuit, especially the mover head. From the overlap analysis, it seems the model utilizes single-hop information to conduct reasoning. We discover an intriguing phenomenon in GPT-

Table 5: Hit for different hop

|      | First-hop | Second-hop | Integrate |
|------|-----------|------------|-----------|
| node | 83.33     | 70.27      | 71.42     |
| edge | 63.20     | 45.27      | 49.42     |

2 and TinyLLAMA: the models can correctly answer first-hop knowledge and perform multi-hop reasoning based on it. However, interestingly, when we delete the first-hop knowledge while retaining only the second-hop relation and the first-hop subject, the models can still correctly answer the multi-hop question. Figure 10 illustrates a specific case in our findings. It further enhances our previous findings that the model actually conducts the factual recall with the relation head and the gathered information about the subject. **Moreover, we found this phenomenon is hugely alleviated by the aligned model, which requires further investigation in the future.**

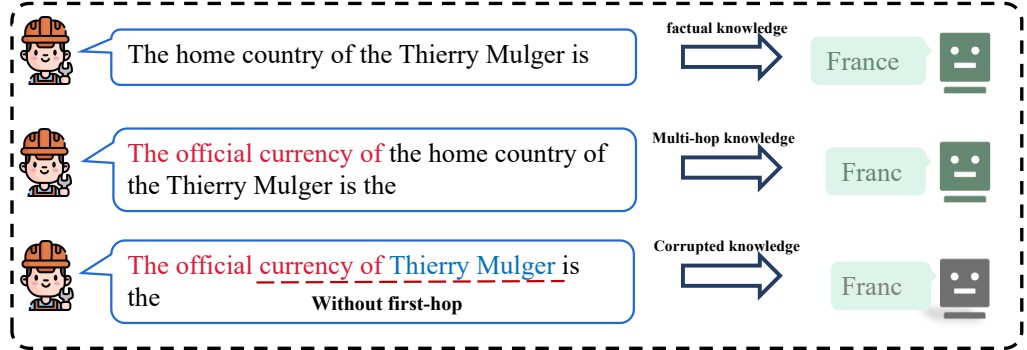

Figure 10: a specific case in Multi-hop reasoning. When we removed the context of the first hop question, we found the model also directly gave us the answer. The phenomenon appears in both GPT2 and TinyLLAMA.

## C.2 Hallucination

The results are presented in Figure 5. We observe an interesting phenomenon: the correct answer and the wrong answer are both accumulated at the previous layer, but at some specific layers, the wrong answer is selected as the answer to extract. We hypothesize that there may be a **circuit competition** here proposed by [72] and we detect the behavior of the specific component between them.

## C.3 Reverse Relation

Reverse Curse [73] is an important issue in language models when models successfully give us the correct answer $o$ for $(s, r)$, but with a reverse relation $\hat{r}$ and $o$, the models fail to give us the correct subject $s$. In this part, we endeavor to investigate how the language model manages the reverse relation when they successfully store the knowledge. We first sample facts where the language model successfully predicts the given knowledge and the reversed fact. Then, we compute the overlap between these two circuits, $\mathcal{C}_d$ and $\mathcal{C}_r$ under node levels based on equation 4. We select the *"superhero_person"* relation to see the difference between these two circuits and test the node overlap of the two circuits in the model. We notice that the overlap between the two circuits is about 70%, indicating the language model may store the related information in the same place. We also found the activated mover heads for the two relationships to be identical.

## D Edit Experiments

### D.1 Method Implementation

**ROME**    As proposed by Meng et al. [18], ROME views knowledge editing as a minimal optimization problem. ROME regards the MLP module as a simple key-value store. Specifically, the key represents a subject and the value encapsulates knowledge about that subject, the MLP can reestablish the association by retrieving the corresponding value for the key. To add a new key-value pair, ROME applies a rank-one modification to the MLP's weights, effectively "writing in" the new information directly. This method enables more direct and precise modification of the model's knowledge. The ROME method for model editing was conducted based on the EasyEdit[74] framework, utilizing the default parameters provided by EasyEdit. The experiments are performed on an A800 80G GPU, with approximately 8GB of memory consumption.

**FT-M**    For Fine-Tuning (FT-M), we follow Zhang et al. [24]. It trains the MLP layer using the cross-entropy loss on the target answer while masking the original text. This approach aligns more closely with the traditional fine-tuning object. The FT-M method is conducted using the EasyEdit[4] [74] framework, with the default parameters provided by EasyEdit. The experiments are also performed on an A800 80G GPU, with a memory consumption of approximately 10GB.

---

[4]https://github.com/zjunlp/EasyEdit

Table 6: Number of examples per relation and the count of accurate predictions by different LMs. This table is borrowed from Hernandez et al. [42] and here we sampled with different ways.

| Category | Relation | # | # Correct in Hit@10 | | |
|---|---|---|---|---|---|
| | | | GPT2-Medium | GPT2-large | TinyLLaMA |
| factual | person mother | 994 | 83 | 144 | 361 |
| | person father | 991 | 359 | 385 | 474 |
| | person sport position | 952 | 400 | 489 | 596 |
| | landmark on continent | 947 | 835 | 543 | 694 |
| | person native language | 919 | 310 | 220 | 260 |
| | landmark in country | 836 | 600 | 489 | 709 |
| | person occupation | 821 | 57 | 76 | 149 |
| | company hq | 674 | 308 | 312 | 470 |
| | product by company | 522 | 422 | 432 | 460 |
| | person plays instrument | 513 | 510 | 505 | 498 |
| | star constellation name | 362 | 266 | 148 | 297 |
| | plays pro sport | 318 | 317 | 316 | 315 |
| | company CEO | 298 | 20 | 52 | 125 |
| | superhero person | 100 | 28 | 35 | 50 |
| | superhero archnemesis | 96 | 6 | 6 | 23 |
| | person university | 91 | 14 | 37 | 35 |
| | pokemon evolution | 44 | 11 | 13 | 16 |
| | country currency | 30 | 25 | 25 | 30 |
| | food from country | 30 | 23 | 25 | 29 |
| | city in country | 27 | 20 | 23 | 27 |
| | country capital city | 24 | 24 | 24 | 24 |
| | country language | 24 | 24 | 24 | 24 |
| | country largest city | 24 | 24 | 24 | 24 |
| | person lead singer of band | 21 | 7 | 16 | 21 |
| | president birth year | 19 | 11 | 12 | - |
| | president election year | 19 | 17 | 18 | - |
| commonsense | object superclass | 76 | 62 | 64 | 72 |
| | word sentiment | 60 | 14 | 9 | 34 |
| | task done by tool | 52 | 44 | 45 | 45 |
| | substance phase of matter | 50 | 12 | 16 | 48 |
| | work location | 38 | 28 | 24 | 27 |
| | fruit inside color | 36 | 36 | 35 | 36 |
| | task person type | 32 | 28 | 27 | 26 |
| | fruit outside color | 30 | 16 | 20 | 21 |
| linguistic | word first letter | 241 | 236 | 235 | 241 |
| | word last letter | 241 | 135 | 73 | 114 |
| | adjective antonym | 100 | 80 | 81 | 84 |
| | adjective superlative | 80 | 24 | 19 | 63 |
| | verb past tense | 76 | 1 | 15 | 76 |
| | adjective comparative | 68 | 7 | 15 | 63 |
| bias | occupation age | 45 | 18 | 20 | 18 |
| | univ degree gender | 38 | 14 | 35 | 38 |
| | name birthplace | 31 | 29 | 30 | 31 |
| | name religion | 31 | 24 | 31 | 31 |
| | characteristic gender | 30 | 26 | 30 | 30 |
| | name gender | 19 | 19 | 19 | 19 |
| | occupation gender | 19 | 19 | 19 | 19 |

## D.2 Edit Cases on FT-M and ROME

When a circuit is established for a particular piece of knowledge, we can manipulate the model's computation by targeting critical points within the circuit. Li et al. [75] ablates a small number of important causal pathways by masking the edges in the circuit and making the model less toxic and safer, which proves the effectiveness of the circuit. As illustrated in figure 12 and figure 13, we present the changes in the ranking of the predicted probabilities for the target new token when editing layer 6, 12, and 18 of the GPT-2 medium model using FT-M and ROME methods. When applying FT-M for model editing, it is evident that the rank of the target new token's probability sharply

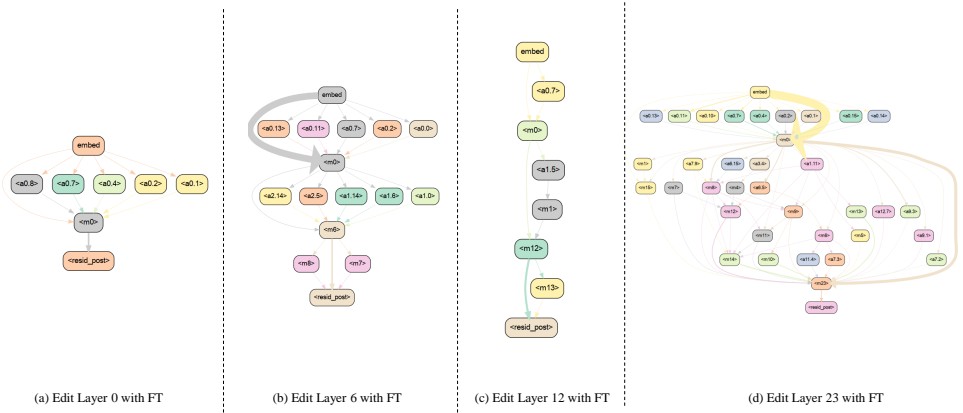

(a) Edit Layer 0 with FT     (b) Edit Layer 6 with FT     (c) Edit Layer 12 with FT     (d) Edit Layer 23 with FT

Figure 11: The knowledge circuit obtained from the edited model for the case *"Platform Controller Hub was created by"* with the target entity "Intel" shows that when editing the model using different layers, the fine-tuned settings allow the edited MLP to directly provide the edited information.

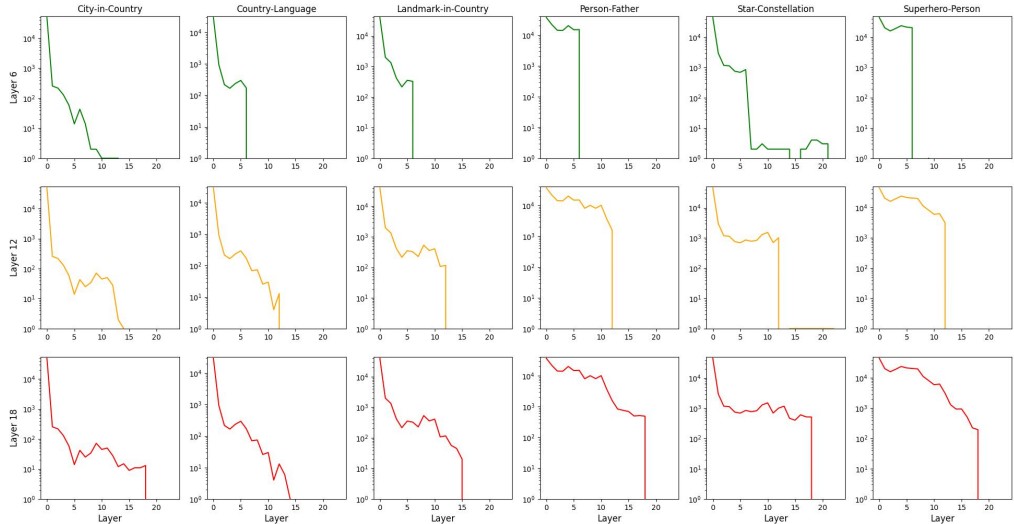

Figure 12: FT-M Rank Change Across Different Layers

declines at the corresponding edited layer, resulting in a vertical line in the figure. This indicates that FT-M directly embeds the editing information into the model's information flow. Conversely, when using the ROME method for editing, this effect is mitigated. The predicted probability of the target new token reaches its peak only a few layers after the edited layer. This observation is consistent with our previous analysis in Section 5.

# E    More related Work and Discussion

## E.1    More Related Work

**Knowledge in MLP**    Geva et al. [76] claim that MLP serves as key-value memories for knowledge in LMs. Geva et al. [77] further propose the knowledge neuron theory, suggesting that the key and

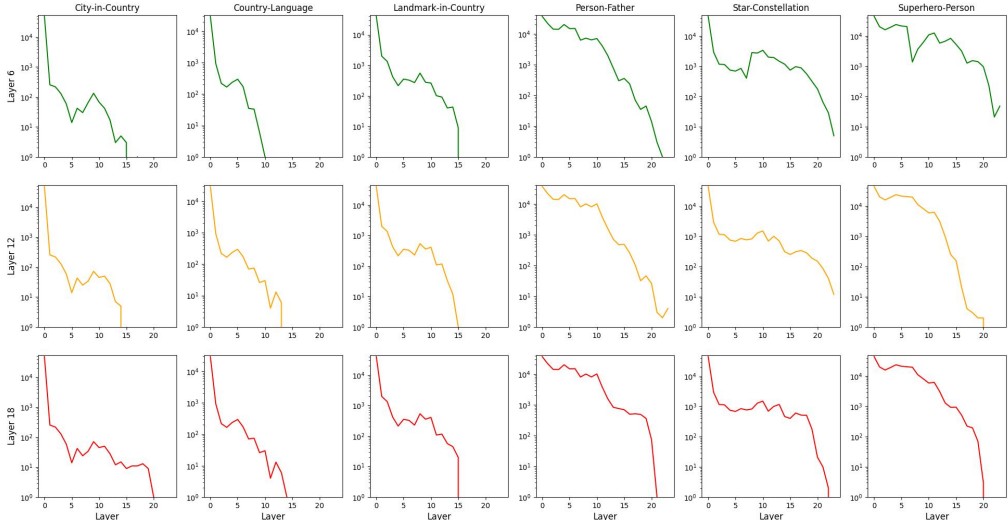

Figure 13: ROME Rank Change Across Different Layers

value vectors in MLPs encode factual knowledge. Based on the above findings, Chen et al. [78] observe that multiple distinct sets of KNs can store identical facts. Chen et al. [79] explore the structural and functional among neurons by neurological topology clustering method. Meng et al. [18] and Meng et al. [80] confirm that MLP modules do store factual knowledge and pioneering use of knowledge editing methods to modify outdated knowledge stored in language models. Anthropic recently introduces scaling monosemanticity[5], which extracts highly abstract features that respond to and behaviorally cause abstract behaviors.

**Knowledge in Attention Heads**    Li et al. [58] reveal that some attention heads are capable of truthful answers. Wu et al. [81] investigate 4 model families, 6 model scales, and 3 types of finetuning, and find retrieval heads, which are responsible for retrieving relevant information from long context. Jin et al. [82] suggest that memory heads can retrieve knowledge from internal memory, while context heads can recall knowledge from external context. Todd et al. [83] use causal mediation analysis on a diverse range of in-context-learning and find some attention heads, dubbed function vectors, which trigger the ability of in-context-learning.

**Knowledge in Hybrid Components**    Recent works emphasize the importance of connections of components among language models for knowledge representation and utilization. Geva et al. [13] describe factual recall by the following three steps: (1) subject enrichment in MLP sublayers, akin to ROME [18], (2) propagation of relations to the END token, and (3) selective extraction of attributes by attention heads in later layers. Lv et al. [31] apply projection and intervention to explore mechanisms in factual recalls tasks and conclude that task-specific attention head may move the topic entity to the final position of the residual stream, while MLP conducts relation function.

**Circuit**    Circuit discovery also plays a significant role in analyzing the internal mechanisms of the entire model [25]. Specifically, a circuit, comprising components such as MLP and attention, is a subgraph of the computation graph. Conmy et al. [32] design an automated circuit discovery approach that implements the specified behavior. Wang et al. [26] explain the circuits for the Indirect Object Identification (IOI) task. They use causal interventions to discover circuits responsible for the flow of information. Instead the above task-specific circuit, Merullo et al. [27] presents evidence a circuit is shared by similar tasks in IOI and Color Object (CO) tasks. Dutta et al. [84] construct circuits using attention heads, and further observe that attention heads are pivotal to chain-of-thought reasoning, i.e., attention heads that move information along ontological relations exclusively appear in the initial half of the layers, while the tokens responsible for writing the answer predominantly appear in the later half of the layers. However, the above-mentioned circuit studies either focus solely

---

[5]https://transformer-circuits.pub/2024/scaling-monosemanticity/index.html

on a single component (MLP or attention) or only explore IOI and CO tasks. IOI and CO tasks necessitate the model to search the preceding context for a matching token and then copy it into the next token prediction. Also, despite the success of previous circuit discovery, we can hardly make it into real usage. Hence, in this work, we attempt to analyze a knowledge circuit consisting of both MLP and attention components and investigate the effect of current editing methods on the circuit to shed light on the future.

**Tools** There are also many tools that are designed to analyze the LM's behavior, such as Logit Lens [85], Attention len [86], Attribution lens [42] and transformer-lens [41]. NeuroX [87] implements various interpretation methods under a unified API and provides insights into how knowledge is structured in representations and discovers the role of neurons in LM. Transformer Debugger [88] is an interpretability tool provided by OpenAI, which deploys the GPT-4 and sparse auto-encoder to explain the language neurons and attention head. PatchScope [89] is a tool provided by Google that uses a new model to explain the hidden states in the original model.

## E.2 Limitation and Future Discussion

Despite of the attempt to combine the attention head and MLP to view the knowledge storage as a whole, this work operates with a relatively coarse granularity of circuits. For instance, the neurons within an MLP may necessitate a finer level of granularity to fully capture their behavior and contributions. Even though we now know these components work together to express the knowledge, why they are activated is still opaque. Our methodology employs the logit lens as a means to detect and analyze component information. However, this approach may encounter discrepancies between the middle layers and the output unembedding matrix. Such discrepancies can hinder a comprehensive and concrete analysis of the circuit components' behavior in the early layers. This limitation suggests the need for more robust techniques to bridge the gap between intermediate representations and final outputs. Recently, the Attention Lens method [86] has been proposed, which involves training a specific unembedding matrix to map each attention head into the vocabulary space. While this method is promising, it is also resource-intensive. Nevertheless, it represents a potential starting point for a deeper understanding of the knowledge circuits within neural models. Moreover, our research indicates that several mover heads are reused across different types of knowledge or relational contexts. The mechanisms by which these heads are activated and the conditions under which they operate require further exploration and may shed light on why neurons are sometimes "monosemantic" responding to a single feature, and sometimes "polysemantic" [90] responding to many unrelated features.

