# OpenReview forum: "Knowledge Circuits in Pretrained Transformers"
_NeurIPS.cc/2024/Conference — NeurIPS 2024 poster_

### Official Review · Reviewer_8n8x · 2024-07-02

**Soundness:** 3
**Presentation:** 2
**Contribution:** 2
**Rating:** 5
**Confidence:** 3

**Summary:**

This paper tries to study how language models (LMs) recall factual knowledge by finding circuits within the model internals in a mechanistic way, on a dataset of (subject, relation object)-like sentences. The discovered circuits help understand the mechanisms of LM's knowledge recall, and also enable some explorations on the impact of knowledge editing methods and analyze hallucinations and in-context learning (ICL).

**Strengths:**

- Discovering the internal mechanisms of LMs' factual recall is an important problem for better understanding and improving LM's knowledge-related abilities. The methodologies and the discovered circuits presented in the paper are interesting and may be useful for the community.
- The analysis based on the discovered circuits is informative and could inspire further applications of circuit discovery.

**Weaknesses:**

- Overall, the findings in this paper are not particularly surprising, and mostly verify known mechanisms studied by prior investigations on knowledge recall in LMs. This could also be seen from the paper writing - in most (if not all) places where a discovered circuit component has a dedicated functionality, it is something already found in the literature (e.g., mover head, relation head, mixture head, etc.). Also given the simplicity of the task (recall the object entity given the subject and relation), it is not hard to anticipate these mechanisms in a transformer model in the first place.

- The introduced methodology is not very technically involved. The adopted circuit discovery method is a direct application of standard causal mediation analysis in the literature, and very much follows the framework of Conmy et al. as cited.

- The interpretations of the mechanisms are somewhat subjective and not rigorously justified. For example, there are usually direct claims about a certain head or a certain MLP layer functioning as such and such, without enough supporting evidence. It would be better to include more statistics to help convince the reader why things are the way they are described in the main text. For example, how to quantitatively define what a "mover head" is, and how did you get this from the statistical evidence?

- There are few in-depth discussions, potentially due to the simplicity of the task. But even within the task itself, the overall findings are quite fragile and it is very hard to get the bigger picture from all the presented findings. Ultimately, the important question is: are there issues with the current LM's mechanisms on factual recall, and how to address them? Take hallucination for example, is it always the case that "the model fails to move the correct knowledge to the final token in the earlier layers"? If this is the case, why does this happen, and is there a general method to resolve the issue besides checking every error example and fixing the problematic components?

- The writing overall could be improved. For example, "zero ablation" and "MatchNLL"  are repeatedly mentioned in Section 3.1 and 3.3. The caption around Figure 2 is not informative enough to understand the figure content (e.g., are these attention/mlp nodes?) and is too close to the main text.

**Questions:**

See weaknesses.

**Limitations:**

Yes.

---

> ### Author Rebuttal · Authors · 2024-08-07
>
> Dear reviewer 8n8x,
>
> Many thanks for your detailed and constructive comments. We hope that this reply answers all your questions, and we look forward to further discussion.
>
> **Addressing Weaknesses and Questions:**
>
> > **WK1&WK2: The findings in this paper are not particularly surprising, and the introduced methodology is not very technically involved.**
>
> - Factual recall may be a simple task for humans, but for current LMs, it’s still challenging, and we still do not know the mechanism of current LMs. In the previous work aiming to understand the behavior of  LM, they mainly focused on the identified objection identification (IOI), which doesn't require memory of the knowledge in the model.  Instead, in our work, we focus on knowledge recall, which is the basis of many complex tasks like multi-hop reasoning and chain-of-thought. As you mentioned in the strengths, our work could inspire further applications of circuit discovery.
>
> - Second, previous findings about the special attention head are intriguing but do not apply to the factual recall task.
> Also, even if we anticipate these mechanisms in a transformer model in the first place, our work provides a new perspective to find them and investigate their behavior change. The analysis through the knowledge editing task enables us to see the mechanism more clearly and understand the drawbacks of the current LM and editing methods, which are also our main contributions.
>
> > **WK3: include more statistics to help convince the reader why things are the way they are described in the main text. For example, how do quantitatively define what a "mover head" is, and how did you get this from the statistical evidence.**
>
> Thanks for your constructive advice. We provide the definition of these heads in Appendix and we will refine it in the main text. Here, we show more concrete definitions and statistics.
> - **Mover Head:** Attention Heads that attend strongly to the subject token and extract attributes of the subject. From Figure 3 and Figure 4, we can see that the rank of the token is similar to the subject position and the last token position. And in the knowledge editing task, we can view the changes in the behaviors.
>
> - **Relation Head:**  Relation heads extract many relation attributes by attending from the end token to the relation tokens.
>
> We select these attention heads by the attention score of the specific direction in which they put more than half of their influence on the attention pattern. To better see the function of these components, we corrupt them in the computation to see the change in the model’s output. The amount of the mover head and relation head we removed is 10.
>
> | probability | Original Model | Ablating Mover Head | Ablating Relation Head |
> | --- | --- | --- | --- |
> | Target token  | 73.71 | 39.31 | 16.32 |
> | Other relation-related token  | 7.2 | 30.85 | 3.7 |
>
> From the Table, we can find the model's performance drop when we remove these special attention heads. Specially, mover head contribute more to the subject-specific information, as ablating them would increase other relation-related tokens' probability.  Moreover, if we ablate the relation head, we can find the model tends to generate some meaningless tokens instead of the relation information.
> We also show one case of the output in Figure 3 in the attached pdf. From the figure, we can find ablating the mover head would increase the probability of "Italian", "English" and "Spanish", which are not subject-related. While ablating the relation head would lead to the increase of some meaningless word "a", "that", which are not relation-related.
>
> > **WK4: Are there issues with the current LM's mechanisms on factual recall, and how to address them? Take hallucinations for example, is it always the case that "the model fails to move the correct knowledge to the final token in the earlier layers"? If this is the case, why does this happen, and is there a general method to resolve the issue besides checking every error example and fixing the problematic components?**
>
> A good question, and that is exactly our ultimate goal. Current LMs still show some disadvantages, and understanding the mechanism is important for us to solve them. In our work, as mentioned by reviewer jmbU and in Figure 8 in the paper, we found the failure in the multi-hop questions scenario can be attributed to saturated information and the skip of the first hop.  Also, regarding the hallucination scenarios when the knowledge is stored in the knowledge model, we can see the mover's head select the wrong information, and until the last layer, the true answer moves to the top token.
>
> For the possible solutions of these issues, knowledge editing currently shows the potential to address hallucinations and errors in large language model. In our analysis, we can see the ROME methods can alter the behavior of the mover head to help it gain the right information. In the future we can conduct some regularization here to make the information here at a proper level.
> As you mentioned, our work could inspire further applications of circuit discovery, We acknowledge that there are still many mysteries that we haven’t uncovered, but we believe our work is a good beginning.
>
> > **WK5: The writing overall could be improved.**
>
> Thanks for your advice. We will polish our paper to make it more clear and easy to understand.

---

> ### Comment · Reviewer_8n8x · 2024-08-11
>
> Thank you for the response, which addresses my concerns to a certain degree. Regarding "previous findings about the special attention head are intriguing but do not apply to the factual recall task", I believe this is not accurate. For example, "Summing Up the Facts: Additive Mechanisms Behind Factual Recall in LLMs" by Chughtai et al (also cited as [15] in the paper here) investigates factual recall and finds "subject heads", "relation heads", 'mixed heads', etc. which overlap a lot with the findings here.
>
> Overall, I believe the major weaknesses I mentioned (lack of novelty in techniques and findings, lack of deeper insights) still hold, but I'll slightly raise my evaluation due to the extra positive signals and promises from the response.

---

> > ### Author Response · Authors · 2024-08-12
> >
> > Thank you for your thoughtful feedback and for considering the additional information provided in our response.
> > We appreciate your reference to Chughtai et al.'s work, “Summing Up the Facts: Additive Mechanisms Behind Factual Recall in LLMs” (Sumup for short). Both our works aims to discover the knowledge mechanism in language model. We agree that there are connections between the identified attention head by Chughtai et al. and our own findings. We will revise our manuscript to better acknowledge these connections and to ensure that we accurately represent the existing literature.
> >
> > Here, we put an extra comparison between our two works:
> > - Our research focuses on identifying the neural circuit within the model that is responsible for knowledge retention. Our experiments show that the identified circuit significantly contributes to the model’s performance. By examining the components within the discovered subgraph, we have identified key attention heads and their interconnections that are crucial to the circuit’s function. In contrast, Sumup directly investigates the behavior of attention heads, with its primary contribution being the identification of these heads. Despite of their claim of the additive mechanism for the factual recall, they simply mentioned in Section 4 in their paper as a speculation. In our work, we can view these connection through the circuit we discovered. Additionally, the additive motif the Sumup paper proposed is not validated. Our experiments suggest that the behavior of the subject mover head changes when the connection to the relation head is removed, indicating potential causal relationships beyond simple additivity.
> >
> > - Our work also adds value by offering a clearer explanation of the model’s operational behavior. Through the circuit perspective, we gain insights into the reasons behind failures in knowledge editing tasks involving locality and reasoning. Furthermore, examining these circuits allows us to understand the model’s behavior in scenarios involving hallucination and in-context learning.
> >
> > We are grateful for your willingness to slightly raise your evaluation based on the positive signals and promises from our response. Regarding the major weaknesses you identified, we understand your perspective, as the circuits we have discovered indeed hold great potential for further investigation. As you noted in your review, our analysis based on the discovered circuits is informative and could inspire further applications of circuit discovery. We believe our work represents a solid foundation in this area and are confident that the revisions we are undertaking will enhance the quality of our paper and more effectively address the concerns you have raised.
> >
> > Thank you once again for your valuable input. We are committed to producing a high-quality, impactful research paper, and we appreciate your role in helping us achieve that goal.
> >
> > Sincerely!

---

### Official Review · Reviewer_8w5y · 2024-07-02

**Soundness:** 4
**Presentation:** 4
**Contribution:** 4
**Rating:** 7
**Confidence:** 4

**Summary:**

The paper introduce a critical subgraph named Knowledge Circuit in the language model to understand how LLMs store and express the knowledge. Through extensive experiments and cases analysis, the authors show the significance of this proposed Knowledge Circuit, which can unveil implicit neural knowledge representations, elucidate internal mechanisms for knowledge editing, and facilitate interpreting language model behaviors.

**Strengths:**

1. The authors approach the knowledge mechanism of LLMs from the Knowledge Circuit perspective, which has not been explored in this field and has good interpretability.

2. The authors conduct extensive experimental analysis, and find that the Knowledge Circuit is beneficial to unveiling implicit neural knowledge representations, understanding knowledge editing mechanism, and interpreting language model behaviors.

3. The paper is written-well, with a direct motivation and a well organized structure.

**Weaknesses:**

1. It would be better if the authors analyze more knowledge editing methods.

2. What is the average size of this subgraph? And what is the distribution of nodes?

**Questions:**

Please see above Weaknesses.

---

> ### Author Rebuttal · Authors · 2024-08-07
>
> Dear reviewer 8w5y,
>
> Many thanks for your detailed and constructive comments. We appreciate that you are optimistic about the impact of the work.
>
> **Response to weakness and questions:**
>
> > **WK: It would be better if the authors analyzed more knowledge editing methods.**
>
> Thanks for your advice, we agree with your opinion. In fact, we previously evaluated the Knowledge Neuron methods to edit and we found utilizing knowledge neurons always failed to overturn the model’s prediction. We would consider other hyper-network editing methods in the future.
>
> > **Question: circuit size and distribution.**
>
> - In Table 1, we show the number of edges of the discovered circuit. Take GPT2-Medium as an example; the average number of edges in the final identified circuit is around 400, and the average number of nodes is about 80.  This proves that with only less than 5% of the original computing subgraph edges, the model can maintain over 70% of its original performance, demonstrating the effectiveness of our methods.
> - We list the distribution of the circuit in Figure 2. Consistent with our expectations, the initial layers of the network, where the model processes inputs and extracts foundational information, are dominated by the activity of attention and feed-forward blocks. As the model progresses to undertake more nuanced and specialized tasks, the engagement of individual attention heads and FFN blocks diminishes correspondingly.

---

> > ### Comment · Reviewer_8w5y · 2024-08-13
> >
> > Many thanks to the authors for the detailed answers. I maintain the positive rating for the paper.

---

### Official Review · Reviewer_7LFR · 2024-07-10

**Soundness:** 2
**Presentation:** 3
**Contribution:** 3
**Rating:** 5
**Confidence:** 4

**Summary:**

This paper focus on the problem of interpreting the knowledge storage mechanism of the large language models. The authors proposed a new perspective, which uses the knowledge circuit to understand how the language model stores and expresses the knowledge. Knowledge circuit is a subgraph of the computation graph of the model. Specifically, the knowledge circuit of a given factual knowledge is defined as the smallest subgraph to recall it "from the knowledge stored in the model". They proposed a knowledge circuit construction method, where the nodes are ablated one-by-one to measure the impact of removing it from the computation graph. The resulted knowledge circuit will then be a necessary computation graph for maintaining the original inference result. Experiments are conducted across a variety of domains of factual recall tasks. To intuitively interpret the constructed knowledge circuits, they probes the information flow by projecting the latent representations into the space of token embeddings. Under this interpretation, several conclusions are drawn as follows.
- language model tends to aggregate knowledge in the earlier to middle layers and further enhances this information in the later layers.
- the knowledge editing method ROME tends to incorporate edited information primarily at the edited layer.
- when hallucination occurs, the language model fails to correctly transfer knowledge to the final token in the earlier layers.
- several new attention heads emerge in the knowledge circuit during in-context learning.

**Strengths:**

- It is in general well formatted and easy to follow.
- Experimental details are well included.

**Weaknesses:**

- Regarding the novelty of this work, reference [36] appears to be highly relevant, as it involves the construction of a flow graph and the presentation of semantic information flow. However, this reference is not discussed in the related work section.
- The robustness of the knowledge circuit construction method should be addressed. For instance, the results of the construction method should be unique or equivalent under different traversal orders. If not, the solution may not be well-defined.
- The interpretation based on the knowledge circuit lacks clarity. Although several conclusions are drawn in the paper, it is unclear if these conclusions align with existing objective observations on related phenomena, such as in-context learning, and whether they could aid in predicting the model's behavior. The effectiveness of an explanation method should be validated by deriving a conclusion or guidance that can be subsequently verified through observation. This aspect is not clearly demonstrated in the paper.
- It would be beneficial to include formal definitions of mover head, relation head, and mixture head, as these are crucial concepts in this paper.

**Questions:**

- Please refer to the weakness part.
- According to Table 1, the performance drop of Circuit varies across different datasets. Is there any analysis of this phenomenon?

**Limitations:**

Yes

---

> ### Author Rebuttal · Authors · 2024-08-07
>
> Dear reviewer 7LFR,
>
> Many thanks for your detailed and constructive comments. We recognize and agree with the limitations of our work and address the specific comments and improvements here.
>
> **Response to weakness and questions:**
>
> > **WK1: Relation with reference [36]**
>
> Thank you for bringing this up! We would modify our paper and add this missing discussion. The aim of the VISIT aligned with our paper. However, they mainly build the graph by logit lens and by pruning the unactivated neuron, while our methods measure the contribution to the prediction. Also, they mainly focus on the behavior of the specific neuron or block in the Transformer model, ignoring the cooperation of these different components, while our work provides a new perspective. Finally, they simply conduct several case analyses and do not consider further analysis of the background mechanism under knowledge editing and hallucinations. We believe our two papers supplement each other, and employing the fine-grained analysis could help us understand our identified circuit more in the future.
>
> > **WK2: Robustness of the methods.**
>
> Very good point.  Our method is computed based on the reverse-topological order (so the nodes are sorted from output to input) to traverse the computation graph.
> - **We put the different computed circuits based on the topological order and our reverse-topological order in Figure 1 in the attached pdf in the global response.** We can see that if we use the original topological orders, the resulting circuits tend to retain redundant nodes in the initial layers. This occurs because the Transformer processes information layer-wise. If nodes from an earlier layer are ablated first, it disrupts the input to subsequent layers, potentially causing a substantial decline in performance. In contrast, when we apply our reverse-topological order, the ablation of nodes in a later layer does not impact the computations of the preceding layers. **Also, the topological order would omit the later layers as in our experiments in the main text, the model would give us the answer before the last layer, leading to an incomplete circuit.**
> - Meanwhile, in our experiments, we found that **our reverse-topological order also takes less time than the topological order.** By ablating nodes in the later layers at the outset, our method significantly reduces the overall computation time. In comparison, the topological order necessitates computing through all layers until the final stage, which is inherently more time-consuming due to the denser circuits in the later layers.
>
> These results demonstrate the robustness of our methods, and from our view, this point deserves a new paper to discuss the casual effect order when computing the graph and is not in the main scope of our work. Thanks again for your very good point.
>
> > **WK3: lacks clarity.**
>
> Thanks for your advice. As to the ICL problem you mentioned, our work supports the previous ICL mechanism [1]. They found the special attention heads’ output acts like the function vector for ICL demonstrations, and if we add the function vector output to the original zero-shot input, the model can do the same task. However, they didn’t analyze the behavior of these attention heads. Based on our proposed methods, we can discover these special attention heads demonstrate the behavior in Figure 5(b).
>
> To clarify our findings, we conduct experiments by ablating the newly appeared attention head in the ICL circuit. **We can find that compared to the randomly selected attention head by ablating this attention, the probability drops significantly in the prediction, demonstrating the importance of the component our method identified.**
>
> |  |  | Origin Model | Ablating extra head in ICL circuit | $\Delta$ (Original - Ablating extra head )| Ablating random head |
> | --- | --- | --- | --- | --- | --- |
> | Linguistic | adj_comparative | 62.24 | 32.55 | 29.69 | 58.18 |
> | Commonsense | word_sentiment | 89.02 | 55.50 | 33.52 | 88.61 |
> |  | substance_phase | 78.74 | 52.85 | 25.89 | 71.24 |
> | Bias | occupation_gender | 86.97 | 59.54 | 27.03 | 86.54 |
> | Factual | person_occupation | 35.17 | 23.27 | 11.90 | 31.60 |
>
> Moreover, for the editing task, we provide another observation of the mover head’s behavior change in Figure 2 in the attached pdf in the global response, which also make our findings more clear. We can find that before editing, the mover head retrieves the correct information for the Windows Server 2003 “Microsoft”, but after the edit, the mover head retrieves the error information ‘Intel’, which is actually the edited information at the subject place by the ROME method.
>
>
> > **WK4: formal definitions of mover head, relation head**
>
> Thanks for your advice. In our previous version, we defined these concepts in the Appendix due to the space limit, and we would mention them in the main paper when modifying it.
>
> > **Question: the performance drop of the circuit varies across different datasets.**
>
> Our experiments test the Hit@10 to measure whether the model knows the knowledge. As we ablate everything but the circuit, we anticipate a little drop-down in the prediction. In our analysis, we found that for some kind of knowledge like "fruit inside color", the original rank of the target token is around 6-8, and after ablating other things, the rank is slightly higher than 10. Hence, this drop-down here is a bit more. The most interesting finding is that we can see an increase in several kinds of knowledge, suggesting that the discovered knowledge circuits may encapsulate the relevant knowledge and the model’s performance on these tasks could have been hindered by noise from other components.
>
> ---
>
> [1] Eric Todd, Millicent L. Li, Arnab Sen Sharma, Aaron Mueller, Byron C. Wallace, David Bau. Function Vectors in Large Language Models (2024)

---

> > ### Comment · Reviewer_7LFR · 2024-08-12
> >
> > I would like to thank the authors for their efforts in addressing my concerns. However, a few issues still persist.
> > Regarding the robustness of the knowledge circuit construction method, the authors acknowledge that the resulting circuits differ depending on the traversal order used. This raises concerns about the lack of rigorous definitions for the term "knowledge circuit" and the absence of an objective metric to evaluate the constructed circuits. The results in Table 1 indicate that there can be significant discrepancies between the decisions of the original model and those produced by the constructed circuit. This brings into question whether the constructed circuits truly align with the original "knowledge."

---

> > > ### Author Response · Authors · 2024-08-13
> > >
> > > Dear reviewer 7LFR,
> > >
> > > We extend our heartfelt gratitude for your invaluable feedback. We hope that our response will serve to address your concerns effectively.
> > >
> > > - Our objective is to identify “knowledge circuits,” which are the sparse computational subgraphs within the Transformer model that encapsulate specific facets of its behavior related to factual recall. In light of the Transformer model’s understanding and the directed acyclic graph (DAG) nature of its computation, we have adopted a reverse-topological approach. This method, which involves tracing information in a top-down manner through the network, has been widely utilized in prior research [1-5] and has consistently demonstrated its reliability and efficacy.Regarding the DAG and activation patching, examining variables in reverse order significantly facilitates the isolation of each variable’s effects, thereby preventing confounding influences from upstream variables—a critical aspect when estimating direct causal effects.
> > > As we mentioned in our previous response, other order tends to maintain some redundant components and omit the later layers in the model, leading to an incomplete graph. It also requires more computation time (in our experiments about 5~8 times lower than the reverse-order).
> > >
> > > - To assess the validity of the constructed circuits, we have employed the circuit-standalone performance metric, a standard approach utilized in earlier studies [1,3-7]. Regarding the performance of Table1, we can see that the circuit-standalone can improve the performance on the test set, suggesting that the discovered knowledge circuits may encapsulate the relevant knowledge and the model’s performance on these tasks could have been hindered by noise from other unrelated components.
> > > To rigorously evaluate whether the constructed circuits genuinely represent the original “knowledge,” we have conducted a comparative analysis among the discovered circuit  $C$, random circuit $C_{random}$ and the complementary circuit $\mathcal{G}-C$ on the test sets. The random circuit matches the size of our discovered circuit, while the complementary circuit represents the graph from which the discovered circuit has been ablated.
> > > |  |  $C_{random}$ | $C$ | $\mathcal{G}-C$ |
> > > | --- | --- | --- | --- |
> > > | Adj Antonym | 0 | 0.40 | 0.00 |
> > > | object superclass | 0 | 0.52 | 0.14 |
> > > | Landmark country | 0 |  0.36 | 0.10 |
> > > | Person Native Language | 0 |  0.76 | 0.17 |
> > > | name birthplace | 0 | 0.57 | 0.05 |
> > >
> > > We can vew that our discovered circuit contribute most to the knowledge recall, indicating the circuit contains most of the nodes related to the knowledge recall.
> > >
> > > We agree that the circuit-discovery method still have room for improvement and we discuss it in the limitations in the paper.
> > > In the recent days, some new works to improve the quality of the circuit emerging [6,8] and we still can see potential to update the circuit discovery through these works.
> > > We believe this is an ongoing research endeavor and we want to underscore that the goal of our work is to find the knowledge circuit in LLM and adopt the circuit to analyze the reason behind of the shortcomings of the current knowledge editing methods and guide the further manipulation of the LM. Our work make a step here and we are committed to continuing this journey.
> > >
> > > Once again, we thank you for your insightful advice and constructive feedback.
> > >
> > > ---
> > > [1] Towards Automated Circuit Discovery for Mechanistic Interpretability.  (2023)
> > >
> > > [2] Representation Engineering: A Top-Down Approach to AI Transparency. (2023)
> > >
> > > [3] Interpretability in the wild: a circuit for indirect object identification in GPT-2 small. (2023)
> > >
> > > [4] Circuit component reuse across tasks in transformer language models. (2024)
> > >
> > > [5] LLM Circuit Analyses Are Consistent Across Training and Scale. (2024)
> > >
> > > [6] Finding Transformer Circuits with Edge Pruning. (2024)
> > >
> > > [7] Have Faith in Faithfulness: Going Beyond Circuit Overlap When Finding Model Mechanisms. (2024)
> > >
> > > [8] Functional Faithfulness in the Wild: Circuit Discovery with Differentiable Computation Graph Pruning. (2024)

---

> > > > ### Comment · Reviewer_7LFR · 2024-08-14
> > > >
> > > > I would like to thank the authors for their thoughtful response. Their willingness to improve the paper moves my opinion to the positive side. I hope they will incorporate the discussed improvements into the final version.

---

> > > > > ### Author Response · Authors · 2024-08-14
> > > > >
> > > > > We deeply appreciate your time and effort in reviewing our work and your support in raising your score. Your assessment has been invaluable in refining our work and clarifying the key aspects of our research. We are committed to refining our experimental approach and analysis in the revised manuscript to reflect the valuable feedback you have provided.
> > > > >
> > > > > We sincerely thank you for your continued time and thoughtful consideration.
> > > > >
> > > > > Warm regards!

---

### Official Review · Reviewer_jmbU · 2024-07-13

**Soundness:** 4
**Presentation:** 4
**Contribution:** 4
**Rating:** 8
**Confidence:** 4

**Summary:**

The authors propose utilizing techniques from mechanistic interpretability to explore the connections between model components involved in factual recall. They systematically ablate connections between model components in a reverse topological order, maintaining a list of the connections that most significantly recover loss on factual recall tasks. Several properties of the resultant "knowledge circuit" are measured, including its sufficiency for recovering model performance on factual recall.

The authors then demonstrate the roles of various components of the knowledge circuit, such as relation heads, mover heads, extract heads, and MLPs. They interpret the roles of these components by projecting their outputs directly into the logit space of the model. This approach is applied to understanding hallucinations and model editing failures.

The authors demonstrate how knowledge editing methods like ROME switch the role of the subject mover head to instead extract the final answer directly from the subject token rather than moving information about the subject.  The authors also manage to explain failures in multi-hop knowledge editing - the edited knowledge "dominantly saturates" the subject mover head preventing it from performing its natural function.  They also show how factual hallucination failures can be attributed to a mistake on the part of the subject mover head.

**Strengths:**

- As far as I am aware, this paper is the first to provide a mechanistic interpretation of the impact of knowledge editing techniques on a model.  The authors demonstrate how knowledge editing methods like ROME switch the role of the subject mover head to instead extract the final answer directly from the subject token rather than moving information about the subject.  The authors also manage to explain failures in multi-hop knowledge editing - the edited knowledge "dominantly saturates" the subject mover head preventing it from performing its natural function.  This work will be quite relevant to other researchers thinking about improving model editing techniques.
- The authors provide a new perspective on the origins of hallucinations in the model. They attribute hallucination failures to the inability of the subject mover head to move the correct answer to the final token position. This mechanistic understanding offers valuable insights into why hallucinations occur and how they might be mitigated.
- The authors contextualize their findings amongst other papers that attempt to mechanistically interpret factual recall in models.  The authors also reproduce a claim mentioned in several previous works, involving components in the early layers of a model being more important for knowledge retrieval (Figure 2).

**Weaknesses:**

- The authors choose to perform ROME [1] edits at layer 0 of the model.  In contrast, the original ROME paper performs edits in the middle of the model.  The precise layer is chosen using a causal tracing algorithm.  This choice is not justified in the paper.  It is unclear what difference this makes in the results provided in this paper.
- In table 1, it is unclear why ablating everything but the circuit improves performance on some tasks, but decreases others.  For example, restricting the model to just the circuit increases the accuracy of "Country Language" but decreases the accuracy of "fruit inside color".

[1] Kevin Meng, David Bau, Alex Andonian, Yonatan Belinkov. "Locating and Editing Factual Associations in GPT." (2023).

**Questions:**

- Why did the authors choose to perform ROME edits at layer 0 of the model, in contrast to the original ROME paper's approach of editing in the middle layers using a causal tracing algorithm, and what impact might this choice have on the results presented in the paper?
- In Table 1, what explains the inconsistent effects of ablating everything but the circuit on task performance, such as improving accuracy for "Country Language" while decreasing it for "fruit inside color"?

**Limitations:**

The authors discuss the limitations of their work, and how it can contribute to ensuring safety and privacy information to promote trustworthy AI.

---

> ### Author Rebuttal · Authors · 2024-08-07
>
> Dear reviewer jmbU,
>
> We are very happy that you are optimistic about the impact of the work. In this comment, we respond to the question and the suggested improvements.
>
> **Response to weakness & questions:**
>
> > **WK1&Q1: About the editing layer we select in our experiments.**
>
> The layer in current editing methods is an important hyper-parameter; currently, there is no conclusion on which layer is the best.  In ROME[1] paper, they edit layer 0 for the FT method and in our experiments, we follow their setting as the case to compare the behaviors of ROME and FT under the same setting.
>
> Moreover, we **also conduct editing experiments under different layers in Appendix, Figure 11 and 12**. We found the same phenomenon across these different layers, which enhances our previous findings. Actually, despite the casual tracing results pointing out that the middle layer stores the knowledge, the identified layer is not always the best layer to edit. In Figure 12, editing ROME using the mid-layer 6 failed to overturn the model's results for the ‘Super-hero’ knowledge. This phenomenon is also found in previous work[2]. Our work provides a new view of what happened when we edited the knowledge.
>
> > **WK2&Q2: inconsistent effects of ablating everything but the circuit.**
>
> Our experiments test the Hit@10 to measure whether the model knows the knowledge. As we ablate everything but the circuit, we anticipate a little drop-down in the prediction.
> In our analysis, we found that for "fruit inside color",  the original rank of the target token is around 6-8, and after ablating other things, the rank is slightly higher than 10.
> Commonly, the drop-down is acceptable and the most interesting finding is that we can see an increase in several kinds of knowledge, suggesting that the discovered knowledge circuits may encapsulate the relevant knowledge and the model’s performance on these tasks could have been hindered by noise from other components.
>
> ---
>
> [1] Kevin Meng, David Bau, Alex Andonian, Yonatan Belinkov. "Locating and Editing Factual Associations in GPT." (2023).
>
> [2] Peter Hase, Mohit Bansal, Been Kim, Asma Ghandeharioun. “Does Localization Inform Editing? Surprising Differences in Causality-Based Localization vs. Knowledge Editing in Language Models.” (2023)

---

> > ### Comment · Reviewer_jmbU · 2024-08-12
> > **Response to rebuttal**
> >
> > Thank you for taking the time to respond. I appreciate your thoughtful feedback, which has helped strengthen my analysis of the comparison between ROME editing and FT.  This has increased my confidence in the high quality of your work.

---

### Official Review · Reviewer_Njw6 · 2024-07-13

**Soundness:** 3
**Presentation:** 3
**Contribution:** 3
**Rating:** 8
**Confidence:** 5

**Summary:**

This paper presents a circuit-based analysis of factual recall and model editing in language models. The authors make two key claims about knowledge storage and retrieval: 1) Factual recall is compatible with circuit-based frameworks. 2) Traditional model editing methods influence behavior change within the model's "knowledge circuit." The paper compellingly advocates for studying knowledge storage and recall using the circuit framework, presenting preliminary but promising results.

**Strengths:**

This paper is well-written, and the experiments are presented with clarity. While some of the results are still preliminary, the paper provides a novel perspective on interpreting and understanding the subject.

**Weaknesses:**

## General Comments

Despite the strengths and contributions highlighted, a significant shortcoming of this paper is the absence of a concrete, successful application of circuits for the tasks mentioned in the introduction and Section 6, such as model editing, hallucination detection, and in-context learning (ICL). This is why I described the review as "preliminary," even though the paper provides concrete experiments and evidence on the existence of a connection between factual knowledge recall and circuits.

Despite its shortcomings, this paper offers an intriguing and promising new approach to addressing the problem by building on Niu et al.'s [21] call to "...examine the entire decision-making circuit" when studying how knowledge is stored and recalled in language models (LMs) with supporting evidence and experiments. The authors highlight the limitations of the previous knowledge neuron theory of LMs, noting its inadequacies as a fundamental theory for interpreting the factual recall process in LMs. The community needs a new theoretical framework, and I believe this paper points us in the right direction.

### Knowledge Circuit Information Analysis

This section of the experiment aims to demonstrate Claim #1: that circuits can adequately explain the process of factual storage and recall. While the experimental results provide some preliminary support for this claim, the evidence is not as compelling as suggested in the introduction. Although the model shows initial success in using circuit-based interpretations for factual recall, some of the results are underwhelming. Notably, the relation "capital city" has a test set performance of 0%. Despite the authors' assertion that the standalone performance of the circuit is good, there is no baseline performance for comparison. Perhaps using random circuits as a baseline could provide a clearer benchmark.

### Related Work

The paper provides an adequate summary of prior work in Section 7 and the introduction; however, there is room for improvement. The introduction initially highlights the issue with previous knowledge editing methods at lines 40-42, noting that "different types of knowledge are often located in the same areas." While this is a valid issue, it is not the most critical one. The relevant work section (lines 336-339) offers a more comprehensive summary of the issues. Therefore, it would be beneficial to expand the introduction by incorporating a condensed summary from lines 336-339 to provide a clearer and more thorough context for the readers.

**Questions:**

Please see the weaknesses section.

**Limitations:**

The authors have adequately addressed the limitations.

---

> ### Author Rebuttal · Authors · 2024-08-07
>
> Dear reviewer Njw6,
>
> Thank you very much for your well-considered review! We are very happy that you appreciate the work we put into exploring a novel perspective on interpreting and understanding the subject. Thank you also for the insightful questions, and please let us know if you have any more.
>
> **Response to weakness:**
>
> > **WK1:** **absence of a concrete, successful application of circuits.**
>
> We agree that our current work still has room for improvement in the proposed application. However, as you mentioned, the main contribution of our work is proving the existence of a connection between circuits and these applications, including model editing, hallucination, and ICL. We are also investigating to address these issues and we believe our work is a good start for future work in this field.
>
> > **WK2: no baseline performance for comparison.**  Thanks for your advice. We have conducted experiments that compare the performance of randomly constructed circuits. The results can be seen as follows:
>
> |  | Random Circuit  | Our discovered Circuit |
> | --- | --- | --- |
> | Adj Antonym | 0 | 1.00 |
> | world first letter | 0 | 0.88 |
> | world last letter | 0 | 0.72 |
> | object superclass | 0 | 0.68 |
> | fruit inside color | 0 | 0.20 |
> | work location | 0 | 0.70 |
> | Capital City | 0 | 1.00 |
> | Landmark country | 0 | 0.60 |
> | Country Language | 0 | 1.00 |
> | Person Native Language | 0 | 0.76 |
> | name religion | 0 | 0.50 |
> | occupation age | 0 | 1.00 |
> | occupation gender | 0 | 0.66 |
> | name birthplace | 0 | 0.57 |
>
> Here, we compute the random circuits by randomly deciding whether the edge should be removed and making sure the graph is connected. **The random circuit is the same size as the circuit we discovered by our method.**
> From the table, we can see that the random circuit failed to maintain the model’s performance, which further enhanced the robustness and efficacy of our methods.
>
> Meanwhile, Regarding some odd performances like the "capital city”, we conduct more analysis here. We found the test data here may not be stored in the language model. In this case, even if we find the circuit for the knowledge, the circuit itself still cannot provide us with the correct answer as the LM itself does not store the knowledge in the model.
>
> > **WK3: related work to be improved.**
>
> Thanks for your advice. Due to the limited pages of the submission, we put some of the related work in Appendix E. Your insights are greatly appreciated and will be taken into consideration.  We would polish our introduction to make it more convincing.

---

> > ### Comment · Reviewer_Njw6 · 2024-08-12
> > **Acknowledgement to the rebuttal**
> >
> > Thank you for providing a response to my comments and questions. It is very helpful. I maintain my positive review to the paper.

---

### Author Rebuttal · Authors · 2024-08-07

Dear Reviewers,

We thank you all for the detailed and constructive comments.

*All* reviewers together found our contribution “a novel perspective on interpreting, points us in the right direction.” (Njw6) and “first to provide a mechanistic interpretation of the impact of knowledge editing techniques on a model” (jmbU) that “has not been explored in this field and has good interpretability” (8w5y) and “well formatted and easy to follow”(7LFR); “could inspire further applications of circuit discovery and may be useful for the community” (8n8x).

We sincerely appreciate your positive comments.

We’ve attached a PDF with several figures in this global response to tackle some questions provided by reviewers, and in our individual responses to reviewers, we hope that we have answered specific questions.

Our PDF includes:

1. Figure 1 compares the discovered circuit under reverse-topological order and topological order in GPT2-small under the same setting, demonstrating our methods' robustness. (reviewer 7LFR)
2. Figure 2 shows the behavior change of the edited model by ROME, clarifying our findings. (reviewer 7LFR)
3. Figure 3 shows the output of the original model and the model in which we ablate the mover head and relation head to support their function in a circuit. (reviewer 8n8x)

---

### Decision · Program_Chairs · 2024-09-25

**Decision:**

Accept (poster)

**Comment:**

This paper considers how circuits are instrumental in articulating specific knowledge in LLMs. Experiments on GPT2 and TinyLLAMA allowed observations of how certain model components collaboratively encode knowledge within the model. Further, this paper provides explanations of hallucination and in-context learning via the circuit perspective.

The reviewers consider this paper well-written.

The mechanistic findings about knowledge editing techniques (e.g., the edited knowledge ‘dominantly saturates’ the subject mover head) are among the earliest of its kind. I see two other works of this kind [1,2] but they are contemporaneous and are on different granularity levels — [1] on neuron-level circuits, [2] on SAE-level circuits, and this paper is on module-level circuits.

[1] What do the circuits mean? https://arxiv.org/abs/2406.17241

[2] Sparse feature circuits https://arxiv.org/abs/2403.19647

While the reviewers unanimously agreed on positive scores after discussion, some useful points can help improve the paper. The reviewers recommended some additional elaborations on concepts adopted from past papers along the mech interp thread, e.g., the mover head, relation head, and mixture heads.

Additionally, it would be great to further analyze how the revealed findings (which are somewhat subjective) can benefit the model development. To me, this is a good point but it is a point towards the interpretability research field in general.